# Understanding while Exploring: Semantics-driven Active Mapping

**Liyan Chen** [*1]**, Huangying Zhan** [2]**, Hairong Yin** [3]**, Yi Xu** [2]**, Philippos Mordohai** [1]

[1] Stevens Institute of Technology     [2] Goertek Alpha Labs     [3] Purdue University

## Abstract

Effective robotic autonomy in unknown environments demands proactive exploration and precise understanding of both geometry and semantics. In this paper, we propose ActiveSGM, an active semantic mapping framework designed to predict the informativeness of potential observations before execution. Built upon a 3D Gaussian Splatting (3DGS) mapping backbone, our approach employs semantic and geometric uncertainty quantification, coupled with a sparse semantic representation, to guide exploration. By enabling robots to strategically select the most beneficial viewpoints, ActiveSGM efficiently enhances mapping completeness, accuracy, and robustness to noisy semantic data, ultimately supporting more adaptive scene exploration. Our experiments on the Replica and Matterport3D datasets highlight the effectiveness of ActiveSGM in active semantic mapping tasks.

## 1 Introduction

Mobile robots are expected to play a significant role in human-centered environments, such as warehouses, factories, hospitals, and homes, as well as in dangerous settings, such as mines and nuclear facilities. Rich and accurate geometric and semantic representations are necessary in these scenarios so that robots can understand, interpret, and interact meaningfully with their surroundings. For instance, in automated warehouses, robots are required to recognize various items and place them in the correct sorting zones accordingly. Scene understanding is enabled by a semantic map that is linked to the geometric map [1], which represents the spatial layout of an environment, and enriches it with high-level information such as object categories, surface labels, and functional affordances [2–5]. Such maps are critical for a range of tasks including navigation, inspection, object manipulation, human-robot interaction, and long-term autonomy.

Despite substantial advances in semantic mapping, most current approaches are unable to determine the most informative path for the robot. Instead, they passively rely on externally determined trajectories or predefined exploration strategies [6–9], leading to incomplete or suboptimal scene understanding. In this paper, we present an approach of active semantic mapping that seeks to close the loop between perception and action. This allows agents to plan their next moves and observations to improve the quality, completeness, and efficiency of the semantic map. Our approach, named ActiveSGM (Active Semantic Gaussian Mapping), is the first active semantic mapping system based on radiance fields, enabling rapid exploration, efficient environment understanding, and high-fidelity real-time rendering, ultimately leading to more intelligent and efficient robotic behaviors. ActiveSGM aims to infer the semantic labels of all visible surfaces, without favoring any particular label.

To select the most informative views for the robot, we seek to quantify both geometric and semantic uncertainty. At the geometric level, uncertainty is typically measured by the expected error in the estimated 3D coordinates [10–12]. At the semantic level, uncertainty estimation primarily captures ambiguity among semantic classes. Recent surveys on semantic uncertainty quantification [13–15] found that it is inherently dependent upon the choice of semantic representation.

---

[*]Corresponding author (lchen39@stevens.edu)

39th Conference on Neural Information Processing Systems (NeurIPS 2025).

The choice of semantic representation plays a critical role in semantic mapping systems, which commonly adopt two forms: probability distributions or embeddings. For distribution-based representations, existing methods (e.g., [16–18]) employ either hard or soft assignment strategies. Hard assignments, such as one-hot encoding, strictly assign a single label to each pixel. In contrast, soft assignments allocate a complete categorical probability distribution to each 3D primitive, naturally capturing uncertainty but also incurring higher memory requirements as the number of categories grows. Alternatively, embedding-based representations can also be viewed as a form of soft assignment. Methods like [9, 19] utilize features, such as those from DINO [20] or CLIP [21], to encode semantic embeddings. However, these embeddings are high-dimensional, posing challenges for storage and real-time rendering in large scenes. Consequently, some approaches compress the features into lower-dimensional spaces, such as the three-dimensional RGB color space. The dimension of the embedding feature space directly determines the effectiveness of class discrimination. High-dimensional embeddings, like those from DINO or CLIP, provide a sufficiently expressive feature space to effectively distinguish categories. However, as embeddings become compressed, for instance into RGB space, color blending during multi-view reconstruction inevitably occurs, producing blended colors that may correspond to unrelated categories instead of the original ones.

In this paper, we address semantic representation under the closed-vocabulary assumption, adopting a probability distribution approach that we argue offers better categorical discrimination. In Section 3, we discuss how to store high-dimensional probability distributions within our proposed sparse semantic representation.

To summarize, we propose the first dense active semantic mapping system built upon a 3D Gaussian Splatting (3DGS) backbone, which integrates semantic-aware mapping and planning for active reconstruction. This enables the robot to construct a more accurate geometric map and a richer semantic map with fewer observations. Our method addresses several key challenges:

- **Semantics-aware exploration**: We design a novel semantic exploration criterion that enhances semantic coverage and facilitates disambiguation across observations during exploration.
- **High-dimensional semantic representations and memory footprint**: We adopt a closed-vocabulary setting and introduce a sparse semantic representation that retains the top-$k$ most probable categories, reducing memory overhead without sacrificing semantic richness.
- **Robustness to noisy semantic observations**: Unlike prior works that rely on ground-truth labels, real-world deployment requires handling noisy semantic predictions. We use a pre-trained segmentation model to generate these inputs and design our pipeline to tolerate and progressively refine them, achieving high segmentation quality.

## 2 Related Work

In this section, we review prior work, starting from dense SLAM, active mapping, semantic mapping, and concluding with active semantic mapping. We focus on methods utilizing either Neural Radiance Fields (NeRF) [22–24] or Gaussian Splatting (GS) [25–27] as the representation.

**Dense SLAM.** Autonomous robotics relies on foundational capabilities such as localization, mapping, planning, and motion control [28]. The need to realize these capabilities has spurred advancements in various areas, including visual odometry [29, 30], structure-from-motion (SfM) [31], and Simultaneous Localization and Mapping (SLAM) [32–34, 10, 35]. For surveys of the impact of radiance fields in SLAM and robotics in general, we refer readers to [36–38, 27]. Progress in radiance fields has given rise to a multitude of dense SLAM methods, that estimate depth for almost every pixel of the input images, using NeRF (or other implicit representations, such as TSDF) [39–48] or GS [49–54] to represent scene geometry and appearance. We use SplaTAM [51] as the SLAM backbone of our algorithm.

**Active Mapping.** The goal of SLAM is to estimate the camera/vehicle trajectory from sensor data. Active mapping, or exploration, is a related problem in the domain of active perception [55, 56], where the goal is guiding the sensor to acquire images beneficial to a downstream task. The most common objectives are to reduce uncertainty, equivalently to increase information gain, [57] or to detect and visit frontiers [58]. Prior approaches such as [58–60] adopt a long-term planning paradigm, in which the information gain is estimated along multiple candidate trajectories before execution. Subsequent

studies [61–63], on the other hand, reformulate the problem within a next-best-view framework, where the exploration process is guided by sequential decisions that progressively construct the overall trajectory. Early work demonstrated the effectiveness of active mapping [64–68], while overviews of the state of the art can be found in [10–12].

**Active Mapping using Radiance Fields.** Recently, NeRF-based approaches have been applied to path planning [69] and next-best-view selection [70–72], though they are often limited by their high computational cost [73]. To overcome these limitations, hybrid models such as ActiveRMAP [74] integrate implicit and explicit representations. NARUTO [61] introduces an active neural mapping system with 6DoF movement in unrestricted spaces, while Kuang et al. [73] integrate Voronoi planning to scale exploration to larger environments. 3DGS offers a faster alternative, making real-time mapping and exploration more feasible. Recent works like ActiveSplat [75] utilize a hybrid map with topological abstractions for efficient planning, ActiveGS [76] also uses a hybrid map and associates a confidence with each Gaussian to guide exploration, and AG-SLAM [77] incorporates 3DGS with Fisher Information to balance exploration and localization in complex environments. ActiveGAMER [63] introduces a rendering-based information gain criterion that selects the next-best view for enhancing geometric and photometric reconstruction accuracy in complex environments. RT-GuIDE [78] uses a simple uncertainty measure to achieve real-time planning and exploration on a robot. Recently, NextBestPath [79] considers longer horizons than just the single next view. Like all the methods in this paragraph, however, it does not consider semantics.

**Semantic Mapping.** The goal of semantic mapping is to infer scene descriptions that go beyond geometry [1]. In general, methods in this category endow their 3D representation with semantic labels, which are inferred via semantic segmentation of the input RGB or RGB-D images. Early work includes approaches such SemanticFusion [80], Fusion++ [4], PanopticFusion [81] and Kimera [6], which have adopted different representations exploring tradeoffs between precision and efficiency. Radiance field-based methods are surveyed by Nguyen et al. [82]. Among them, GSNeRF [83] introduces the Semantic Geo-Reasoning and Depth-Guided Visual modules to train a NeRF that encodes semantics along with appearance. Wilson et al. [15] use the variance of the semantic representation at each Gaussian as a proxy for semantic uncertainty. HUGS [84] jointly optimizes geometry, appearance, semantics, and motion using a combination of static and dynamic 3D Gaussians. Logits for all classes are stored with the Gaussians, but the number of classes is small. All of these approaches operate on all frames in batch mode, however.

**Semantic SLAM.** Gaussian splats are well suited for semantic mapping because they can encode additional attributes and are amenable to continual learning, unlike NeRF [85]. All methods below operate on RGB-D video inputs. We point out the important representation choices made by their authors. SGS-SLAM [7] augments a GS-based SLAM system with additional test-time supervision via 2D semantic maps. The authors argue that any off-the-self semantic segmentation algorithm can be integrated in SGS-SLAM and use ground truth labels to supervise the splats for simplicity. High-dimensional semantic labels are converted to "semantic colors" to save space. NIDS-SLAM [86] uses a 2D transformer [87] to estimate keyframe semantics, also converting the semantic labels into "semantic colors". NEDS-SLAM [8] reduces the memory footprint of the high-dimensional semantic features obtained by DINO [20] to three values per splat via a lightweight encoder. OpenGS-SLAM [9] infers consistent labels via the consensus of 2D foundational models across multiple views. It can handle an open vocabulary, but stores only one label per splat.

To overcome the limited dimensionality of colormaps, researchers have endowed the splats with embeddings of the high-dimensional vectors of logits. SNI-SLAM [88] model the correlations among appearance, geometry and semantic features through a cross-attention mechanism and use feature planes [41] to save memory. DNS-SLAM [89] relies on a multi-resolution hash-based feature grid. Optimization is performed in latent space, while ground truth 2D semantic maps are used as inputs. SemGauss-SLAM [90] augments the splats with a 16-channel semantic embedding and presents semantic-informed bundle adjustment. The paper includes results using ground truth 2D labels for supervision, as well as labels inferred by a classifier operating on DINOv2 [91] features. Hier-SLAM [5] addresses the increased storage requirements via a hierarchical tree representation, generated by a large language model. It can handle over 500 semantic classes, but it is also provided the ground truth semantic maps of the images during optimization.

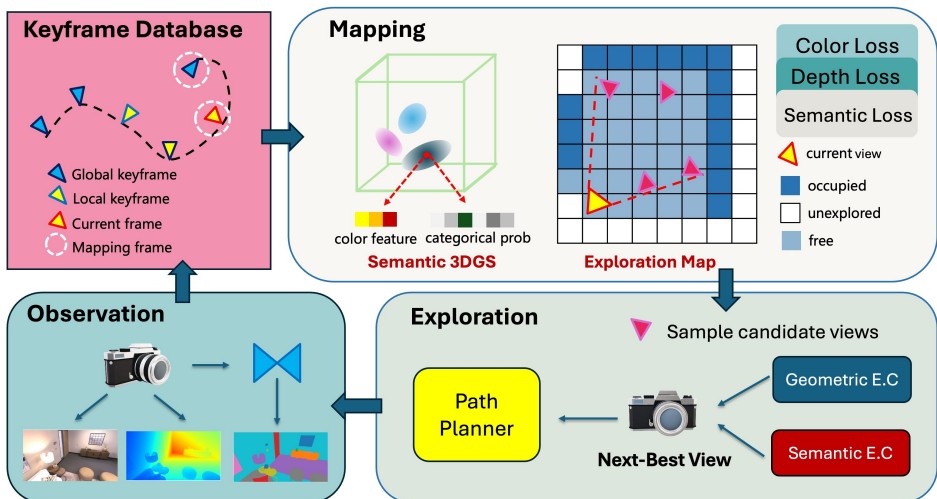

Figure 1: **Overview of the ActiveSGM System.** Our framework integrates observation, mapping, and planning into a unified active semantic mapping system. At each time step, posed RGB-D frames along with semantic predictions from OneFormer [17] are stored in a keyframe database. Selected frames are used to update a Semantic Gaussian Map that encodes geometric, photometric, and semantic properties and is optimized through differentiable rendering. An occupancy-based Exploration Map is updated using the current view and used to sample candidate viewpoints in free space. Next-best views are selected by jointly evaluating geometric and semantic exploration criteria (E.C.), and a path planner navigates toward the selected pose. This closed-loop system enables efficient, semantics-aware reconstruction and exploration in complex 3D environments.

**Active Semantic Mapping.** The above semantic mapping and SLAM approaches are "passive" in the sense that the camera is not actively controlled but follows a predetermined trajectory. In contrast, methods such as [60, 92] actively plan trajectories that maximize the mutual info rmation between past and future semantic observations. Other relevant works have focused on object search using semantic contextual priors—e.g., leveraging knowledge that cups are typically found in kitchens—without explicitly predicting semantic labels for every point in the map [93–95]. Among these approaches, more relevant to ours is the work of Zhang et al. [96] that relies on semantic mutual information and properties of the SLAM pose graph for metric-semantic active mapping. An octree is used to maintain the map, but the current implementation is limited to 2D motion and 8 classes, while ground truth labels are used as semantic observations. Marza et al. [97] added a semantic head to Nerfacto [98] and used it for active mapping of appearance, geometry and semantics. They compared using ground truth semantic labels and Mask-R-CNN [99] to detect 15 object categories, and observed large differences in the metrics. Exploration policies are trained using reinforcement learning and consider the 15 object categories. Unlike our approach, the trajectory is restricted to the ground plane. It is not clear how this approach would have to be modified if all semantic classes in the scene would have to be considered.

## 3 Method

In this section, we present Active Semantic Gaussian Mapping (ActiveSGM), a 3D Semantic Gaussian Splatting framework for active reconstruction that tightly integrates semantic-aware mapping and planning. Section 3.1 introduces Semantic Gaussian Mapping, an efficient representation that enables high-fidelity geometric, photometric, and semantic reconstruction. To reduce the computational and memory overhead of semantic mapping, we propose a sparse semantic representation that supports efficient storage and fast rendering. Building on this, Section 3.2 describes our exploration strategy for next-best-view selection, which jointly leverages geometric and semantic cues to guide the reconstruction of high-quality semantic maps. We outline the ActiveSGM framework in Figure 1.

## 3.1 Semantic Gaussian Mapping

**Gaussian Mapping.** Gaussian Mapping leverages 3DGS to represent scenes as collections of 3D Gaussians, encoding both appearance and geometry for real-time rendering of high-fidelity color and depth images. Building upon the work of Kerbl et al. [25], we adopt the streamlined approach proposed in SplaTAM [51]. This method employs isotropic Gaussians with view-independent color, optimizing parameters such as color ($\mathbf{c}$), center position ($\boldsymbol{\mu}$), radius ($r$), and opacity ($o$).

A notable advantage of 3DGS is its capability for real-time rendering, enabling the synthesis of high-fidelity color and depth images from arbitrary camera poses. This is achieved by transforming 3D Gaussians into camera space, sorting them front-to-back, projecting them onto the 2D image plane, and employing alpha-blending for compositing. The color, depth, and silhouette at pixel $\mathbf{p}$ are rendered from the Gaussian map, where the silhouette indicates whether $\mathbf{p}$ receives a significant projection from any Gaussian. The general rendering process is formulated as

$$R(\mathbf{p}) = \sum_{i=1}^{n} z_i f_i(\mathbf{p}) \prod_{j=1}^{i-1} (1 - f_j(\mathbf{p})), \tag{1}$$

where $z_i \in \{c_i, d_i, 1\}$ and $R(\mathbf{p}) \in \{C(\mathbf{p}), D(\mathbf{p}), S(\mathbf{p})\}$ depending on whether color, depth, or silhouette is being rendered. $f_i(\mathbf{p})$ is derived from the Gaussian's position and size in 2D pixel space. The differentiable nature of this rendering process allows for end-to-end optimization, where gradients are computed based on discrepancies between rendered images and RGB-D inputs, and the optimization objective is formulated as:

$$L = \sum_{\mathbf{p}} (S(\mathbf{p}) > 0.99) (L_1(D(\mathbf{p})) + 0.5 L_1(C(\mathbf{p}))), \tag{2}$$

where only pixels inside the silhouette are considered.

**Semantic Prediction.** We incorporate semantics into the 3DGS map by using OneFormer [17], a state-of-the-art model for unified segmentation, to perform semantic segmentation. Its predictions serve as our primary source of semantic observations.

**Sparse Semantic Representation.** Given the semantic predictions from OneFormer, represented as a probability distribution $\mathcal{P} = (p_1, p_2, ..., p_M)$ over $M$ semantic categories, a straightforward approach to constructing a Semantic Gaussian Map is to incorporate $\mathcal{P}$ as an additional attribute in each 3D Gaussian. However, storing and optimizing such high-dimensional semantic properties can lead to significant memory overhead.

To mitigate this issue, we introduce a sparse semantic representation, where only the top-$k$ categories with the highest probabilities from the initial observation are retained per Gaussian. Specifically, for each Gaussian $\mathbf{G}_i$, we define the sparse semantic vector as $\tilde{\mathcal{P}}_i = (p_{i_1}, p_{i_2}, ..., p_{i_k})$. This compact form preserves most of the semantic information while significantly reducing storage and computation costs. As new observations arrive, the probabilities are updated while keeping the original top-$k$ indices fixed, allowing semantic refinement over time without restoring the full distribution.

**Semantic Rendering.** Similar to color and depth, semantic rendering projects 3D Gaussians into 2D and composites their semantic properties at each pixel. To preserve efficiency, we render only using Gaussians within the current view and aggregate their sparse top-$k$ semantic distributions into a full semantic probability map. Given each Gaussian's sparse vector $\tilde{\mathcal{P}}_i$, we compute the class-$m$ probability at pixel $\mathbf{p}$ as:

$$\mathcal{P}_m(\mathbf{p}) = \sum_{i=1}^{n} p_{i,m} f_i(\mathbf{p}) \prod_{j=1}^{i-1} (1 - f_j(\mathbf{p})), \tag{3}$$

where $p_{i,m}$ is the probability of class $m$ for Gaussian $\mathbf{G}_i$, and $f_i(\mathbf{p})$ denotes its projected influence at pixel $\mathbf{p}$. This approach enables smooth, class-wise semantic rendering while avoiding the overhead of fully dense representations, striking a balance between accuracy and memory efficiency.

**Semantic Loss.** To optimize the semantic 3DGS, we employ a combination of the Hellinger distance and cosine similarity losses. Predictions from OneFormer serve as the pseudo-ground truth $\mathcal{P}_{\text{GT}}$, while the rendered semantic outputs are treated as predictions $\mathcal{P}_{\text{pred}}$. To filter out uncertain supervision, we apply an entropy-based mask $M_H = \mathbb{I}(H(\mathbf{p}) < \tau)$ on $\mathcal{P}_{\text{GT}}$, where $\mathbb{I}(\cdot)$ is the indicator function and $\tau$ is the entropy of a uniform distribution over $k$ categories, i.e. $\tau = \log(k)$. The entropy at each pixel is computed as:

$$H(\mathbf{p}) = -\sum_{m=1}^{M} \mathcal{P}_m(\mathbf{p}) \cdot \log \mathcal{P}_m(\mathbf{p}). \tag{4}$$

The Hellinger distance encourages the predicted semantic distribution to closely match the pseudo ground truth while providing smooth and bounded gradients. To further regularize the optimization, we incorporate the cosine similarity loss, which promotes angular alignment between the predicted and target distributions. This combination ensures both probabilistic accuracy and structural consistency, leading to more stable and robust training for semantic 3DGS. The final semantic loss is defined as:

$$L_{\text{seman}} = M_H \cdot \left( \lambda_{\text{HD}} D_{\text{HD}}(\mathcal{P}_{\text{GT}} \, \| \, \mathcal{P}_{\text{pred}}) + \lambda_{\text{cos}} \left( 1 - \cos(\mathcal{P}_{\text{GT}}, \mathcal{P}_{\text{pred}}) \right) \right), \tag{5}$$

where $D_{\text{HD}}(\cdot \, \| \, \cdot)$ denotes the Hellinger distance and $\cos(\cdot, \cdot)$ is the cosine similarity. We set $\lambda_{\text{HD}} = 0.8$ and $\lambda_{\text{cos}} = 0.2$ to balance their contributions.

To prevent noisy semantic predictions from affecting the entire 3DGS representation, we restrict backpropagation of this loss to only the semantic attributes of each Gaussian, leaving geometric and photometric components untouched.

**Keyframe Selection Strategy.** Following SplaTAM [51], our Gaussian Mapping backbone optimizes the map using a subset of keyframes instead of all input frames. Every fifth frame is considered a keyframe candidate, and the map is updated using local keyframes with the highest 3D overlap, computed by backprojecting depth maps and evaluating visibility within keyframe frustums. This provides efficient multiview supervision but may overfit occluded regions, reducing opacity for valid Gaussians behind surfaces.

To address this, we introduce a global-local keyframe strategy. In addition to local keyframes, we select global keyframes based on: (1) low rendering quality, and (2) low semantic entropy and fewer unknown labels to ensure confident supervision. These global keyframes help cover under-observed and ambiguous regions. In practice, we maintain a 50-50 mix of local and global keyframes to balance local detail with global coverage.

## 3.2 Exploration Planning

To enable efficient semantic reconstruction, we design an exploration planning module that actively selects informative viewpoints. Each candidate pose is evaluated using two criteria: *geometric coverage*, measured by silhouette completeness, and *semantic uncertainty*, quantified by entropy. These criteria approximate information gain [10, 12], which measures the expected reduction in uncertainty from new observations. While computing true information gain is intractable in high-dimensional semantic maps [100], our entropy- and coverage-based approximations allow efficient real-time scoring of candidate viewpoints. To keep computation efficient, we maintain a dynamic candidate pool and adopt a coarse-to-fine sampling strategy that first explores broadly, then refines with denser sampling. We now detail the geometric and semantic exploration criteria, the overall scoring formulation, and the implementation of candidate management.

**Geometric Exploration Criterion.** We adopt ActiveGAMER's [63] exploration criterion formulation to evaluate the geometric coverage of candidate viewpoints. Given a candidate viewpoint $v$, we compute its exploration criterion $\mathcal{I}_{geo}^{v}$ based on the rendered silhouette $S^v$ with respect to the up-to-date Semantic Gaussian Map. The number of missing pixels in the rendered silhouette, denoted as $N_{S^v}$, quantifies the exploration criterion for the candidate viewpoint, which is formulated as:

$$\mathcal{I}_{geo}^{v} = \sigma(\log(N_{S^v})), \quad N_{S^v} = \sum_{\mathbf{p}} \mathbb{I}(S^v(\mathbf{p}) = 0) \tag{6}$$

where $\sigma(\cdot)$ is the softmax function, which normalizes the scores across all candidate viewpoints, and $\mathbb{I}(\cdot)$ is the indicator function, counting pixels with zero values in the silhouette.

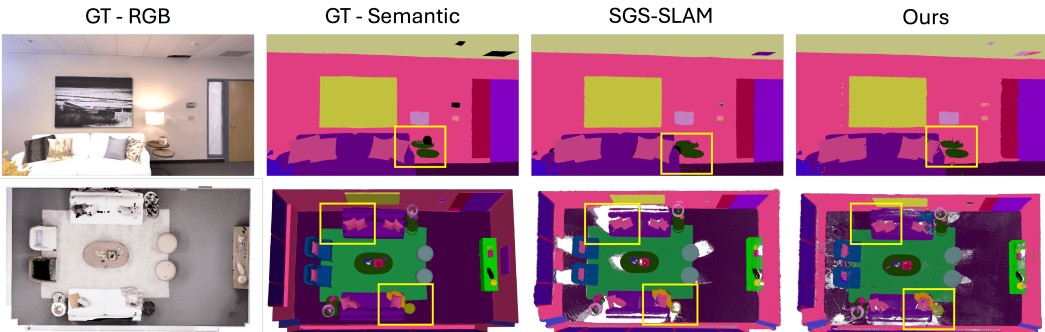

|  GT - RGB | GT - Semantic | SGS-SLAM | Ours |

Figure 2: **Qualitative Results for Replica.** Our method generates denser and more accurate semantic maps than SGS-SLAM, with fewer exploration steps. Yellow boxes highlight improved boundaries and semantic consistency. Black regions denote unknown labels.

**Semantic Exploration Criterion.** In addition to geometric coverage, we assess semantic uncertainty by rendering the semantic probability map of each candidate viewpoint from the current Semantic Gaussian Map. To ensure numerical stability, we clip the probabilities to $[0.001, 1]$ and normalize them to form valid probability distributions. Given a candidate pose $v$, the semantic exploration score is defined as:

$$\mathcal{I}^v_{\text{seman}} = \sigma \left( \sum_{\mathbf{p}} H^v(\mathbf{p}) \right),\tag{7}$$

where $H^v(\mathbf{p})$ is the entropy at pixel $\mathbf{p}$, computed as in Eqn. 4. This encourages selecting views that reduce semantic uncertainty and improve coverage in ambiguous regions.

**Overall Exploration Criterion.** To guide efficient scene coverage and reduce redundant motion, we define the overall exploration criterion by combining geometric and semantic objectives with a motion cost that penalizes distant candidate viewpoints. This encourages the system to prioritize informative poses that are also close to the current camera location.

Given a candidate camera pose $v$, we first compute the exploration criteria $\mathcal{I}^v_{\text{geo}}$ and $\mathcal{I}^v_{\text{seman}}$ as described above. To encourage travel efficiency, we define a motion cost based on the $L_2$ distance between the candidate pose location $T^v_x$ and the current camera location $T^t_x$, denoted as $l^v = \|T^v_x - T^t_x\|_2$. We apply a softmax function to the motion cost to normalize the cost across all candidates. The final distance-aware exploration criterion is defined as:

$$\mathcal{I}^v = (1 - \sigma(l^v)) \cdot \left( \mathcal{I}^v_{\text{geo}} \cdot \mathcal{I}^v_{\text{seman}} \right).\tag{8}$$

This formulation balances information gain and travel efficiency, favoring views that improve map quality while minimizing unnecessary motion.

**Exploration Strategy.** To efficiently evaluate candidate viewpoints, we maintain an *Exploration Map*, a voxel-based occupancy grid that tracks free space. Newly observed voxels are identified by comparing the updated grid to its previous state, and candidate viewpoints are sampled from these new voxels. Candidate positions are spaced every $v_1$ units of length, with $v_2$ viewing directions uniformly distributed using the Fibonacci lattice. Each pose $T^v$ is scored using the overall exploration criterion (Eqn. 8), and low-value candidates ($N_{S_i} < 0.5\%$ of image pixels) are pruned from the pool.

To balance speed and coverage, we use a coarse-to-fine strategy: the coarse stage samples on a single height plane with larger steps ($v_1 = 1$) and fewer directions ($v_2 = 5$); the fine stage increases density with smaller steps ($v_1 = 0.5$), multiple heights, and more directions ($v_2 = 15$), removing redundant views to maintain exploration efficiency and completeness.

## 4 Experiments and Results

### 4.1 Experimental Setup

In this paper, we focus on evaluating the semantic segmentation accuracy of the proposed pipeline, particularly under noisy observation settings, which better reflect real-world conditions. Deploying

and evaluating on physical robots is challenging due to the lack of well-annotated data—especially ground-truth semantic labels- for reliable evaluation. Consequently, prior methods are typically evaluated in simulation, making photorealistic environments the most practical and fair testbed for benchmarking. Following the common evaluation protocol, all of the following experiments are conducted in a simulated environment.

**Simulator and Datasets.** We use Habitat [101] to generate RGB-D frames and OneFormer [17] for semantic segmentation. Frames are captured at $680 \times 1200$ resolution with $60°$ vertical and $90°$ horizontal FOV. The Exploration Map uses a voxel size of 5 cm.

We evaluate on three photorealistic datasets: **Replica** [102], **ReplicaSLAM**, and **MP3D** [103]. Replica includes high-fidelity meshes and 101 semantic classes; we use 8 scenes from [44]. ReplicaSLAM provides predefined camera trajectories for the same 8 scenes. MP3D includes 40 semantic classes; we use 5 scenes for evaluation. Each experiment runs for 2,000 steps on Replica and 5,000 on MP3D, with early termination if the exploration candidate pool is exhausted.

**Semantic Model Fine-tuning.** To improve semantic prediction accuracy, we collect 500 RGB-Semantic frames from each scene and fine-tune OneFormer separately on Replica and MP3D. The fine-tuned models are used to generate per-pixel semantic class probability maps, which are then converted into sparse semantic representations for each 3D Gaussian.

**Semantic Evaluation Metrics.** We follow SGS-SLAM's [7] evaluation protocol and compute the *Average Mean Intersection over Union (mIoU)* by mapping the rendered semantic predictions to ground-truth categories *within each test view*. In addition, we evaluate per-pixel semantic classification using *Top-1* and *Top-3 Accuracy*, and assess the *complete* category distribution (not limited to the categories present in a given image) using *Mean Average Precision (mAP)* and *F1-score*. Owing to space constraints, we report only a subset of the results in Table 1; please refer to Section S.2 of the supplementary material for the full results.

**Geometric and Photometric Metrics.** We evaluate geometric reconstruction using three metrics: *Accuracy* (cm), *Completeness* (cm), and *Completeness ratio* (%) with a 5 cm threshold. These are computed by uniformly sampling 3D points from both the ground-truth mesh and the reconstructed Gaussian Map. For measuring rendering quality, we use *PSNR*, *SSIM*, *LPIPS* and *Depth L1 (D-L1)*.

**Baselines.** To the best of our knowledge, this is the first work to tackle dense active semantic mapping with 3D Gaussian Splatting (3DGS); direct, one-to-one baselines are therefore difficult to establish. We evaluate our system against three categories of baselines: (1) *semantic SLAM methods* based on NeRF or 3DGS, which primarily target segmentation and rendering quality; (2) a *passive semantic mapping pipeline* reconfigured from the strongest representative in (1), to highlight the benefits of our semantic representation and active exploration policy; and (3) *geometry-based active mapping methods*, which focus on maximizing 3D reconstruction accuracy.

All experiments were conducted on two NVIDIA RTX A6000 GPUs. Additional implementation details and extended results are provided in the supplementary material.

## 4.2 Semantic Segmentation Evaluation

**Comparison with NeRF/GS-based Semantic SLAM on ReplicaSLAM.** Most existing NeRF/3DGS-based semantic mapping approaches—such as NIDS-SLAM[86], DNS-SLAM[89], SNI-SLAM[88], and SGS-SLAM[7]—are formulated as semantic SLAM systems, i.e., they jointly solve for localization and semantic mapping. We report results from SGS-SLAM [7] only for reference and completeness since we assume perfect localization. We follow the SGS-SLAM evaluation protocol, which measures the agreement between rendered semantic masks and ground-truth labels visible in each view (Table 1 yellow ). Our setup differs from prior baselines in three key aspects: (1) we use pseudo labels generated by OneFormer [17] instead of ground truth; (2) we evaluate on views not seen during training; and (3) we train with only one-third of the available images. Despite these constraints, our method achieves performance comparable to fully supervised baselines by effectively fusing noisy predictions across views into a coherent semantic map.

Table 1: **Semantic Segmentation Results.** We evaluate ActiveSGM on Replica and MP3D without access to ground-truth semantic labels, requiring fewer mapping steps, and testing on novel views not seen during training. We report average mIoU on Replica and average IoU on MP3D. "GT" means ground-truth labels. while "Pred." means predicted labels from OneFormer.

| Method | Dataset | Labels | Evaluation View | Steps ↓ | Avg. [m]IoU (%) ↑ | F-1 (%) ↑ |
|---|---|---|---|---|---|---|
| NIDS-SLAM [86] | ReplicaSLAM | GT | Train | 2000 | 82.37 | – |
| DNS-SLAM [89] | ReplicaSLAM | GT | Train | 2000 | 84.77 | – |
| SNI-SLAM [88] | ReplicaSLAM | GT | Train | 2000 | 87.41 | – |
| SGS-SLAM [7] | ReplicaSLAM | GT | Train | 2000 | **92.72** | – |
| OneFormer [17] | ReplicaSLAM | GT | Novel | 3000 | 65.41 | – |
| **Ours** | ReplicaSLAM | Pred. | Novel | **713** | 85.13 | – |
| SGS-SLAM [7] | Replica | Pred. | Novel | 2000 | 80.42 | 18.70 |
| Ours (Passive) | Replica | Pred. | Novel | 2000 | 80.14 | 67.81 |
| **Ours** | Replica | Pred. | Novel | **777** | **84.89** | **77.56** |
| SSMI [92] | MP3D | GT | Train | – | 36.14 | – |
| TARE [104] | MP3D | GT | Train | – | 31.70 | – |
| Zhang et al. [96] | MP3D | GT | Train | – | 42.92 | – |
| **Ours** | MP3D | Pred. | Novel | – | **65.58** | – |

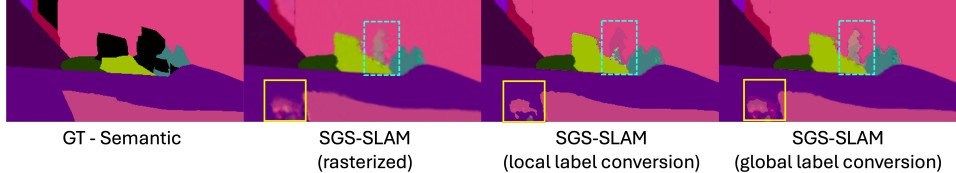

| GT - Semantic | SGS-SLAM (rasterized) | SGS-SLAM (local label conversion) | SGS-SLAM (global label conversion) |

Figure 3: **Color-Coding Ambiguities.** SGS-SLAM blend colors leading to label confusion, especially under global conversion, and the introduction of irrelevant categories.

**Comparison with Passive Semantic Mapping on Replica (Novel Views).** To enable a fairer comparison with semantic SLAM methods, we select SGS-SLAM as the strongest representative baseline and reconfigure it to function as a semantic mapping-only system: (1) We disable its tracking module and use ground truth poses. (2) Replace ground truth semantics with semantic predictions. This allows us to directly evaluate the semantic representation quality, which is a key contribution of our method. We then disable our own active exploration module to ensure a passive, mapping-only setting, enabling a fair apples-to-apples comparison. The results in Table 1 `blue` show that our full pipeline with active exploration achieves better segmentation with fewer observations/steps, demonstrating the effectiveness of our exploration strategy guided by semantic and geometric uncertainty.

**Color-Coding Limitations.** As shown in Figure 3, SGS-SLAM and similar methods use color encoding to represent semantic labels, which often blend during multi-view fusion and introduce arbitrary labels, leading to misclassification and ambiguity (yellow boxes). To recover labels, they apply nearest-color matching using either *local label conversion*, which maps to the nearest color among ground-truth classes in the current view, or *global label conversion*, which considers all ground-truth classes in the scene. However, assuming access to view-specific ground-truth labels is unrealistic. Inconsistencies between local and global conversion are shown in cyan boxes.

**Comparison with Active Semantic Mapping on MP3D.** We also include comparisons to recent active semantic mapping baselines, which are more aligned with our task definition. These further validate the advantage of our sparse semantic representation and active policy. We evaluate on 5 large indoor scenes from MP3D (Table 1 `red`). Table 1 shows active semantic mapping baselines [92, 104] from [96]. We do not know which scenes were used by the baselines, but we evaluate on a common set

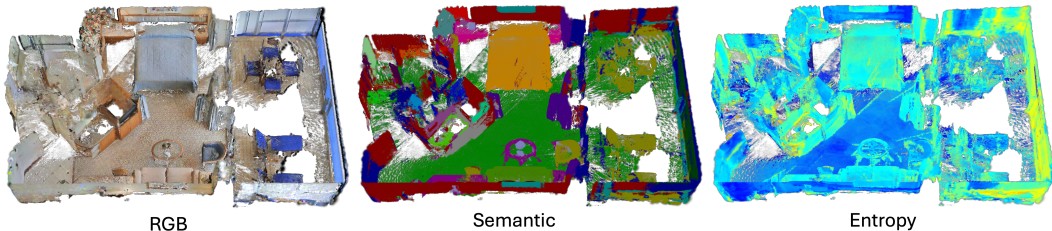

| RGB | Semantic | Entropy |

Figure 4: **Qualitative Results for MP3D.** Top-down visualizations of reconstructed scene, semantic labels and semantic entropy heatmap (low, high). Notably, our results show no high-entropy regions, and produce coherent and dense semantic reconstructions even in large scale MP3D scenes.

Table 2: **Ablation of Semantic Components.** Experiments on *office0* and *room0* from Replica to evaluate the impact of the number of retained categories (*Top-k*) and the use of Hellinger distance (*H.D.*), KL-Divergence (*KL.*) and cosine similarity (*Cos.*) in the semantic loss.

| Top-*k* | H.D. | KL. | Cos. | Avg. mIoU (%) ↑ | Top-1 Acc (%) ↑ | Top-3 Acc (%) ↑ | mAP (%) ↑ | F-1 (%) ↑ |
|---------|------|-----|------|------------------|------------------|------------------|-----------|-----------|
| Top-5 | ✓ | | ✓ | 83.06 | 95.66 | 99.68 | 94.79 | 74.24 |
| Top-8 | ✓ | | ✓ | 83.34 | **95.70** | 99.66 | **95.05** | 74.23 |
| Top-16 | ✓ | | ✓ | **84.08** | 95.68 | **99.73** | 94.92 | **74.73** |
| Top-16 | ✓ | | | 82.26 | 95.57 | 99.61 | 94.21 | 74.10 |
| Top-16 | | | ✓ | 82.70 | 95.62 | 99.73 | 94.40 | 72.43 |
| Top-16 | | ✓ | ✓ | 82.22 | 95.63 | 99.70 | 94.66 | 73.93 |
| Top-16 | ✓ | | ✓ | **84.08** | **95.68** | **99.73** | **94.92** | **74.73** |

of labels, and report Average IoU. All baselines use ground truth labels during optimization. Despite relying on predicted labels and novel views, our method significantly outperforms all baselines. Figure 4 shows that our system produces clean and consistent semantic maps across complex indoor scenes.

**Comparison on 3D Reconstruction and Novel View Synthesis.** We evaluate ActiveSGM 3D reconstruction and novel view synthesis on MP3D and Replica. On MP3D, our method achieves 1.56 cm accuracy and 97.35% completeness, surpassing ActiveGAMER [63] (1.66 cm, 95.32%). In novel view synthesis on Replica, ActiveSGM achieves an SSIM of 0.96, closely matching ActiveGAMER's 0.97 despite not using a photometric refinement stage. This highlights ActiveSGM's ability to maintain a balance between photometric quality and geometric fidelity. Full quantitative and qualitative results are provided in the supplement.

### 4.3 Ablation Studies

We perform ablation studies on two key components of our proposed method that influence semantic mapping performance: (1) the number of categories used in the sparse semantic representation, and (2) the effect of individual loss terms in optimizing semantic features. Experiments are conducted on the *office0* and *room0* scenes from the Replica dataset. As shown in Table 2: (1) Using more categories improves accuracy, we retain the top-16 categories for the best overall performance. (2) Removing either the Hellinger distance or cosine similarity reduces the Average mIoU. Using both terms together, accuracy reaches $84.08\%$, confirming their effectiveness. (3) We also compare KL divergence and Hellinger distance, finding that the Hellinger distance is a more effective choice. Notably, KL divergence can lead to gradient vanishing due to the instability of the logarithmic function, necessitating gradient norm clipping during training.

## 5 Limitations and Conclusion

Despite the strong performance of ActiveSGM we show above, several limitations remain: **Perfect Localization:** We assume known robot poses throughout the process; in real deployments, a separate localization or tracking module would be required. **Perfect Execution:** The robot is assumed to precisely follow planned trajectories; navigation errors should be considered for deployment. **Semantic Segmentation Model:** Our system relies on an external segmentation model (OneFormer) for semantic predictions, which may introduce domain-specific errors. Stronger or fine-tuned models can improve results, while weaker ones may degrade semantic quality. **Limited Joint Optimization:** To stabilize training, we block gradients from the semantic loss to geometric properties, decoupling the optimization of geometry and semantic features.

In conclusion, We presented ActiveSGM, the first dense active semantic mapping system built on a 3D Gaussian Splatting backbone. By unifying geometry, appearance, and semantics, ActiveSGM enables efficient exploration and high-quality mapping with fewer observations. It improves semantic coverage through uncertainty-guided exploration, reduces memory via top-$k$ sparse representations, and handles noisy predictions without ground-truth labels. We hope ActiveSGM will inspire future research in active mapping, semantic understanding, and autonomous robotic exploration. Our code is avaiable at `https://github.com/lly00412/ActiveSGM.git`.

**Acknowledgement.** This research has been supported in part by the National Science Foundation under award 2024653.

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

# Supplement

In this supplement, we provide a detailed outline structured as follows: Section S.1 offers additional implementation details of ActiveSGM. Section S.2 includes extended quantitative and qualitative results, along with a runtime analysis. Section S.3 provides justifications for the checklist items.

## S.1  Implementation Details

**Hardware and Software.** We conducted the experiments on a server with 2 NVIDIA RTX A6000 GPUs and an Intel i9-10900X CPU with 20 cores. Our ActiveSGM is implemented with python 3.8 and CUDA 11.7. Please refer to Section S.3.4 for more information about baselines and other packages we used. Our code can be found at `https://github.com/lly00412/ActiveSGM.git`.

**OneFormer Finetuning Details.** Following the approach of ActiveGAMER [63], we implemented the geometry-based exploration criterion to construct our fine-tuning dataset. Beginning from a random position, the agent performs 500 exploration steps, collecting 500 RGB-Semantic frame pairs per scene. We then fine-tuned OneFormer [17] separately on the collected data from Replica and MP3D, training for 3,000 steps per scene. The Replica dataset has 101 classes, while MP3D has 40. The fine-tuning process follows the official OneFormer tutorial provided by Hugging Face (`https://huggingface.co/docs/transformers/main/en/model_doc/oneformer`). The novel trajectories described in Table 1 of the main paper are used as the test set. These trajectories are distinct from those used for fine-tuning. The train/test Top-1 accuracy is reported in Table S.1.

**Sparse Rendering.** We illustrate the semantic rendering process using our proposed sparse semantic representation (with fewer classes) in Fig. S.1. The overall rendering process proceeds as follows:

```
For each tile:
    For each pixel in the tile:
        For each batch of Gaussians in the frustum:
            Load batch to shared memory # fewer classes decreases loading time
            For each Gaussian in the batch:
                If pixel is affected:
                    Compute contribution (semantic, alpha)
                    Composite with alpha blending # sparse mode needs fewer iters.
                    Early exit if opacity is sufficient
    Write final semantic
```

If our sparse representation is not used, each Gaussian stores a full probability distribution over all classes, and alpha blending of semantic probabilities is performed by iterating over all classes:

```
For each Gaussian G_i in the batch:
    for idx in range(num_classes+1):
        P[idx] += prob[idx] * alpha[idx] * transmittance[idx]
```

where P is the rendered probability distribution of each pixel. This becomes increasingly inefficient when the number of classes is large and many probabilities are near zero. For instance, in the Replica dataset with 101 classes plus one unknown class, this results in 102 iterations per Gaussian.

In contrast, our sparse rendering strategy stores only the Top-$k$ most probable classes per Gaussian ($k$ « number of classes). During rasterization, alpha blending is performed only over these sparse indices:

```
For each Gaussian G_i in the batch:
    indices = topk_indices in G_i
    for idx in indices:
```

Table S.1: **OneFormer Fintuneing Accuracy**

| Dataset | Splits | Avg. | Of0 | Of1 | Of2 | Of3 | Of4 | R0 | R1 | R2 |
|---------|--------|------|------|------|------|------|------|------|------|------|
| Replica [102] | Train | 97.31 | 98.75 | 98.67 | 99.17 | 97.35 | 96.45 | 98.13 | 98.83 | 91.15 |
|  | Test | 89.12 | 89.07 | 71.84 | 92.76 | 93.18 | 91.54 | 87.37 | 92.25 | 94.96 |
|  |  |  | GdvgF | gZ6f7 | HxpKQ | pLe4w | YmJkq |  |  |  |
| MP3D [103] | Train | 93.87 | 94.37 | 94.99 | 93.84 | 95.22 | 90.94 |  |  |  |
|  | Test | 89.77 | 93.68 | 92.58 | 91.21 | 89.52 | 81.86 |  |  |  |

Table S.2: **Semantic Segmentation on ReplicaSLAM.** Rendered semantics are evaluated on 4 scenes using the SGS-SLAM [7] protocol, which compares predictions to ground-truth categories visible per view. Our method uses semantic predictions from OneFormer and is evaluated on novel views.

| Methods | Semantic | View | Avg. Steps ↓ | Avg. mIoU (%) ↑ | R0 (%) | R1 (%) | R2 (%) | Of0 (%) |
|---|---|---|---|---|---|---|---|---|
| NIDS-SLAM [86] | GT | Train | 2000 | 82.37 | 82.45 | 84.08 | 76.99 | 85.94 |
| DNS-SLAM [89] | GT | Train | 2000 | 84.77 | 88.32 | 84.90 | 81.20 | 84.66 |
| SNI-SLAM [88] | GT | Train | 2000 | 87.41 | 88.42 | 87.43 | 86.16 | 87.63 |
| SGS-SLAM [7] | GT | Train | 2000 | 92.72 | 92.95 | 92.91 | 92.10 | 92.90 |
| OneFormer [17] | GT | Novel | 3000 | 65.41 | 69.06 | 65.71 | 67.01 | 59.85 |
| **Ours** | Pred. | Novel | 713 | 85.13 | 84.54 | 85.98 | 85.40 | 84.60 |

```
P[idx] += prob[idx] * alpha[idx] * transmittance[idx]
```

This reduces the number of memory accesses and blending operations without sacrificing semantic fidelity. By reducing the number of stored logits and accessed channels, our sparse representation speeds up both the memory workload, as more Gaussians can be loaded into shared memory, and the Gaussian processing loop, leading to faster semantic rendering. Please refer to Section S.2.2 for a quantitative runtime comparison.

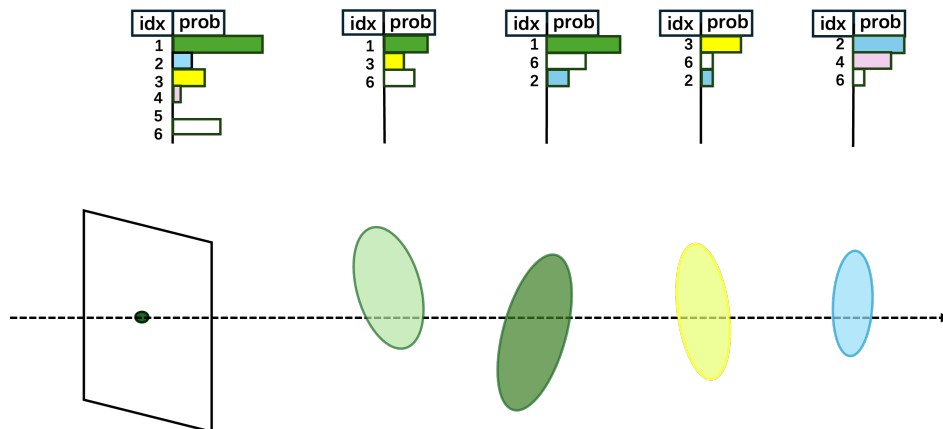

Figure S.1: **Visualization of Rendering Semantic Map with Sparse Semantic Vector.** Each Gaussian only stores indexes and probabilities of the top-$k$ most probable categories, the semantic distribution of the given pixel is rendered following Eqn. (3) in the main paper.

**Local Path Planner.** We employed the Efficient Rapid-exploration Random Tree (RRT) proposed by NARUTO [61] for local path planing. Once the goal location is determined, we use an efficient RRT-based planner to find a path from the current state $s_t$ to the goal $s_g$, using the Exploration Map to measure collision and reachability. (Specifically, the agent should only move within the free voxels defined by the Exploration Map. Additionally, we enforce a collision buffer of 20 cm, ensuring the agent avoids regions that are too close to surrounding surfaces.) To speed up planning in large-scale 3D environments, we enhance standard RRT by also attempting direct connections between samples and the goal. This greatly improves efficiency.

## S.2    Additional Results

### S.2.1    Quantitative Results

**Semantic Segmentation on ReplicaSLAM**  We evaluate on 4 scenes following the SGS-SLAM protocol [7], which compares rendered semantic masks to ground-truth labels visible in each view. The full results are shown in the Table S.2, and have been summarized in Table 1 yellow .

Table S.3: **Semantic Segmentation on Replica (Novel Views).** We present four settings: (1) Fine-tuning results of Oneformer as reference; (2) SGS-SLAM retrained using OneFormer predictions, instead of ground-truth labels as used in Table 1 of the main paper—leads to a noticeable drop in performance; (3) Our method without active exploration, which demonstrates the advantage of the sparse semantic representation alone; (4) Our full pipeline with active exploration, which achieves better segmentation performance with fewer steps.

| Methods | Metrics | Avg. | Of0 | Of1 | Of2 | Of3 | Of4 | R0 | R1 | R2 |
|---|---|---|---|---|---|---|---|---|---|---|
| OneFormer [17] | Steps ↓ | - | - | - | - | - | - | - | - | - |
| | mIoU (%) ↑ | 66.05 | 62.73 | 55.67 | 66.38 | 70.03 | 69.81 | 62.16 | 74.19 | 67.43 |
| | mAP (%) ↑ | 84.59 | 83.29 | 72.47 | 87.39 | 88.36 | 85.83 | 81.56 | 90.68 | 87.12 |
| | F-1 (%) ↑ | 57.96 | 57.78 | 40.45 | 59.51 | 66.33 | 47.89 | 59.20 | 69.70 | 62.81 |
| | Top-1 Acc (%) ↑ | 89.12 | 89.07 | 71.84 | 92.76 | 93.18 | 91.54 | 87.37 | 92.25 | 94.96 |
| | Top-3 Acc (%) ↑ | 96.18 | 96.76 | 89.10 | 96.53 | 97.70 | 97.29 | 96.40 | 96.92 | 98.76 |
| SGS-SLAM [7] | Steps ↓ | 2000 | 2000 | 2000 | 2000 | 2000 | 2000 | 2000 | 2000 | 2000 |
| | mIoU (%)↑ | 80.42 | 77.60 | 75.68 | 78.70 | 78.10 | 89.96 | 83.23 | 83.97 | 76.12 |
| | mAP (%)↑ | 89.94 | 86.37 | 84.67 | 88.63 | 90.98 | 96.22 | 92.10 | 93.80 | 86.73 |
| | F-1 (%)↑ | 18.70 | 18.35 | 15.06 | 19.03 | 17.68 | 18.02 | 25.28 | 18.47 | 17.69 |
| | Top-1 Acc (%) ↑ | 94.42 | 92.68 | 90.06 | 93.52 | 93.42 | 98.14 | 97.16 | 96.71 | 93.64 |
| | Top-3 Acc (%) ↑ | 95.53 | 93.39 | 90.90 | 94.54 | 96.64 | 98.70 | 98.00 | 97.35 | 94.68 |
| Ours (Passive) | Steps ↓ | 2000 | 2000 | 2000 | 2000 | 2000 | 2000 | 2000 | 2000 | 2000 |
| | mIoU(%) ↑ | 80.14 | 74.15 | 74.88 | 76.97 | 79.60 | 88.29 | 84.50 | 84.50 | 78.23 |
| | mAP (%)↑ | 90.09 | 88.91 | 84.86 | 86.27 | 89.12 | 94.86 | 93.78 | 93.78 | 89.13 |
| | F-1 (%)↑ | 67.81 | 64.54 | 53.49 | 72.42 | 64.26 | 66.48 | 70.18 | 80.09 | 71.03 |
| | Top-1 Acc (%) ↑ | 94.05 | 89.80 | 89.69 | 94.99 | 93.83 | 97.60 | 95.65 | 95.65 | 95.16 |
| | Top-3 Acc (%) ↑ | 96.82 | 95.00 | 91.71 | 96.58 | 98.71 | 99.45 | 98.14 | 98.14 | 96.85 |
| **Ours (Active)** | Steps ↓ | 777 | 664 | 501 | 749 | 1175 | 941 | 1082 | 514 | 591 |
| | mIoU (%)↑ | 84.89 | 82.58 | 83.99 | 83.57 | 83.40 | 89.36 | 84.08 | 85.28 | 86.83 |
| | mAP (%)↑ | 94.39 | 94.66 | 91.93 | 92.86 | 93.65 | 96.35 | 95.19 | 94.93 | 95.55 |
| | F-1 (%)↑ | 77.56 | 73.81 | 72.53 | 79.57 | 75.95 | 76.80 | 75.65 | 83.85 | 82.33 |
| | Top-1 Acc (%) ↑ | 96.62 | 94.55 | 96.07 | 98.39 | 94.82 | 97.75 | 96.80 | 96.18 | 98.40 |
| | Top-3 Acc (%) ↑ | 99.52 | 99.76 | 99.01 | 99.77 | 99.51 | 99.58 | 99.69 | 99.05 | 99.81 |

Table S.4: **Semantic Segmentation on MP3D.**

| Methods | Semantic | View | Avg. ↑ | ceilling | appliances | sink | plant | counter | table | mpcat40 |
|---|---|---|---|---|---|---|---|---|---|---|
| SSMI [92] | GT | Train | 36.14 | 46.02 | 41.01 | 25.13 | 39.30 | 36.12 | 29.25 | - |
| TARE [104] | GT | Train | 31.70 | 42.01 | 36.86 | 23.86 | 32.51 | 31.70 | 23.27 | - |
| Zhang et al. [96] | GT | Train | 42.92 | 50.73 | 45.26 | 43.91 | 40.42 | **39.18** | 37.99 | - |
| **Ours** | Pred. | Novel | **65.58** | **70.31** | **76.95** | **69.36** | **73.60** | 14.03 | **69.89** | **55.77** |

**Semantic Segmentation on Replica (Novel Views)** To assess generalization, we generate new trajectories near the SLAM trajectories, following the instructions of SplaTAM [51]. We present the complete results in Table S.3, as a supplement to Table 1 blue in the main paper.

**Semantic Segmentation on MP3D** We also evaluate the average IoU on five large indoor scenes from MP3D (see Table 1 red in the main paper). Table S.4 reports the IoU scores for six common categories, as well as the mean IoU across all 40 categories of our method (denoted as 'mpcat40'). The semantic ground-truth meshes provided by MP3D are noisier than the texture meshes, often containing floaters and missing regions. To ensure a fair comparison, we computed the L1 distance from each point in the semantic mesh to its nearest neighbor in the texture mesh, and filtered out all points with distances greater than 5 cm. Points in the texture meshes inherit the semantic label of the nearest neighboring point in the semantic mesh, if it is within 5 cm, otherwise their labels are set to *unknown*, and then they are used as ground truth in the evaluation. We show an example of the filtered mesh in Figure S.2.

**3D Reconstruction and Novel View Synthesis.** We evaluate the 3D reconstruction and novel view synthesis (NVS) performance of ActiveSGM on MP3D and Replica. The 3D reconstruction results are reported in Table S.5, while the NVS results are presented in Table S.6. Please refer to Section 4.2 for details on how the novel trajectories are generated. Overall, ActiveSGM achieves the best 3D reconstruction and NVS performance on MP3D and performs on par with the state-of-the-art method ActiveGAMER on Replica.

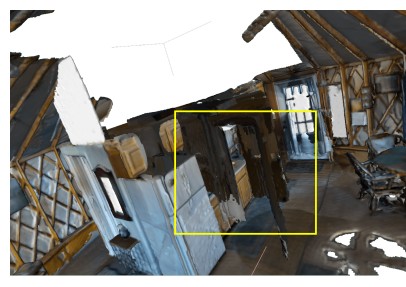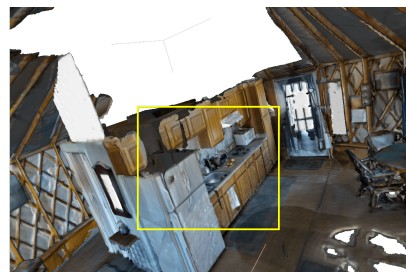

Original                                           Filtered

Figure S.2: **Filtered Semantic Mesh for MP3D.** We present both the original and the filtered semantic mesh from an MP3D scene. After filtering, most of the floaters—such as those highlighted in the yellow box—are successfully removed. The cleaned meshes are then used for semantic segmentation evaluation on MP3D.

Table S.5: **3D Reconstruction Results on Replica and MP3D.** Overall, our method achieves the best performance on MP3D and ranks second on Replica, delivering higher reconstruction accuracy and improved scene completeness compared to prior approaches. Notably, ours is the only method that incorporates semantic information into the exploration criterion, whereas all other baselines rely on geometry-based strategies.

| Methods | Dataset | Acc. (cm) ↓ | Comp. (cm) ↓ | Comp. Ratio (%) ↑ |
|---|---|---|---|---|
| NARUTO [61] | Replica | 1.61 | 1.66 | **97.20** |
| ActiveGAMER [63] | Replica | **1.16** | **1.56** | 96.50 |
| **Ours** | Replica | 1.19 | 1.59 | 96.68 |
| FBE [58] | MP3D | / | 9.78 | 71.18 |
| UPEN [59] | MP3D | / | 10.60 | 69.06 |
| OccAnt [62] | MP3D | / | 9.40 | 71.72 |
| ANM [105] | MP3D | 7.80 | 9.11 | 73.15 |
| NARUTO [61] | MP3D | 6.31 | 3.00 | 90.18 |
| ActiveGAMER [63] | MP3D | 1.66 | 2.30 | 95.32 |
| **Ours** | MP3D | **1.56** | **1.77** | **97.35** |

## S.2.2 Runtime Analysis

We conduct a runtime analysis using the *room0* scene from the Replica dataset to highlight the efficiency of our sparse semantic representation and rendering strategy. The scene, measuring $8\,\mathrm{m} \times 4.8\,\mathrm{m} \times 3\,\mathrm{m}$, is explored and mapped by *ActiveSGM* in 1082 steps over 48 minutes. During the rendering of a semantic map with resolution $(340 \times 600 \times 102)$, approximately 204k Gaussians are involved in the rasterization process. Using a dense semantic representation—where each Gaussian carries a full 102-class probability distribution—the rendering takes 61 ms. In contrast, our sparse semantic representation significantly reduces computation, requiring only 3.1 ms to render the same map. This improvement stems from the reduced number of active channels during rendering and more

Table S.6: **Novel View Rendering Performance on Replica and MP3D.** We report the average rendering metrics across scenes for each method. Our approach delivers consistently strong performance in terms of PSNR, SSIM, LPIPS, and L1 depth error, achieving comparable or better results than baselines, ranking as the second-best on Replica and the best on MP3D. Notably, our method is the only one that also addresses semantic segmentation.

| Method | Dataset | PSNR ↑ | SSIM ↑ | LPIPS ↓ | L1-D ↓ |
|---|---|---|---|---|---|
| SplaTAM [51] | Replica | 29.08 | 0.95 | 0.14 | 1.38 |
| SGS-SLAM [7] | Replica | 27.14 | 0.94 | 0.16 | 7.09 |
| NARUTO [61] | Replica | 26.01 | 0.89 | 0.41 | 9.54 |
| ActiveGAMER [63] | Replica | **32.02** | **0.97** | **0.11** | **1.12** |
| **Ours** | Replica | 30.61 | 0.96 | 0.14 | 1.36 |
| NARUTO [61] | MP3D | 20.52 | 0.72 | 0.58 | 7.95 |
| ActiveGAMER [63] | MP3D | 24.76 | 0.90 | **0.25** | 4.83 |
| **Ours** | MP3D | **26.15** | **0.92** | 0.26 | **3.76** |

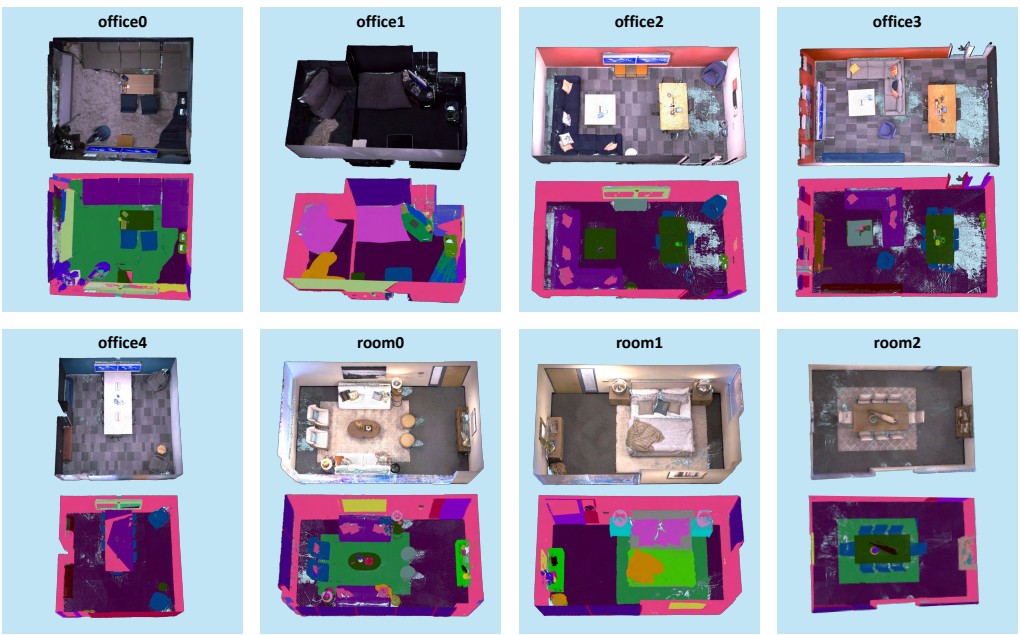

Figure S.3: **RGB and Semantic Reconstruction for Replica.**

importantly from the reduced amount of data transfers on the GPU, showcasing the effectiveness of our sparse approach for real-time semantic mapping.

### S.2.3 Qualitative Results

We also preset the top-down view visualization of the 8 scenes from Replica in Figure S.3 and 5 scenes from MP3D in Figure S.4, please zoom in to see more details.

## S.3 Assets Used and Reproducibility

### S.3.1 Experimental result reproducibility

We provide sufficient implementation details in Section S.1 in the supplement to make results reproducible. We build upon open source software, and our code can be found at `https://github.com/lly00412/ActiveSGM.git`.

### S.3.2 Experimental setting/details

Please refer to Section 4.1 in the main paper.

### S.3.3 Experiments compute resources

Please see the *Hardware* paragraph in Section S.1 in the supplement, and also refer to Sec S.2.2 for runtime analysis.

### S.3.4 Licenses for existing assets

**Datasets.** In this paper, we conduct experiments on the following publicly available datasets. We list the URLs, license information, and citation for each dataset below.

1. **Replica Dataset** [102]
    - URL: `https://github.com/facebookresearch/Replica-Dataset`
    - License: Research or Education only. (`https://github.com/facebookresearch/Replica-Dataset/blob/main/LICENSE`)

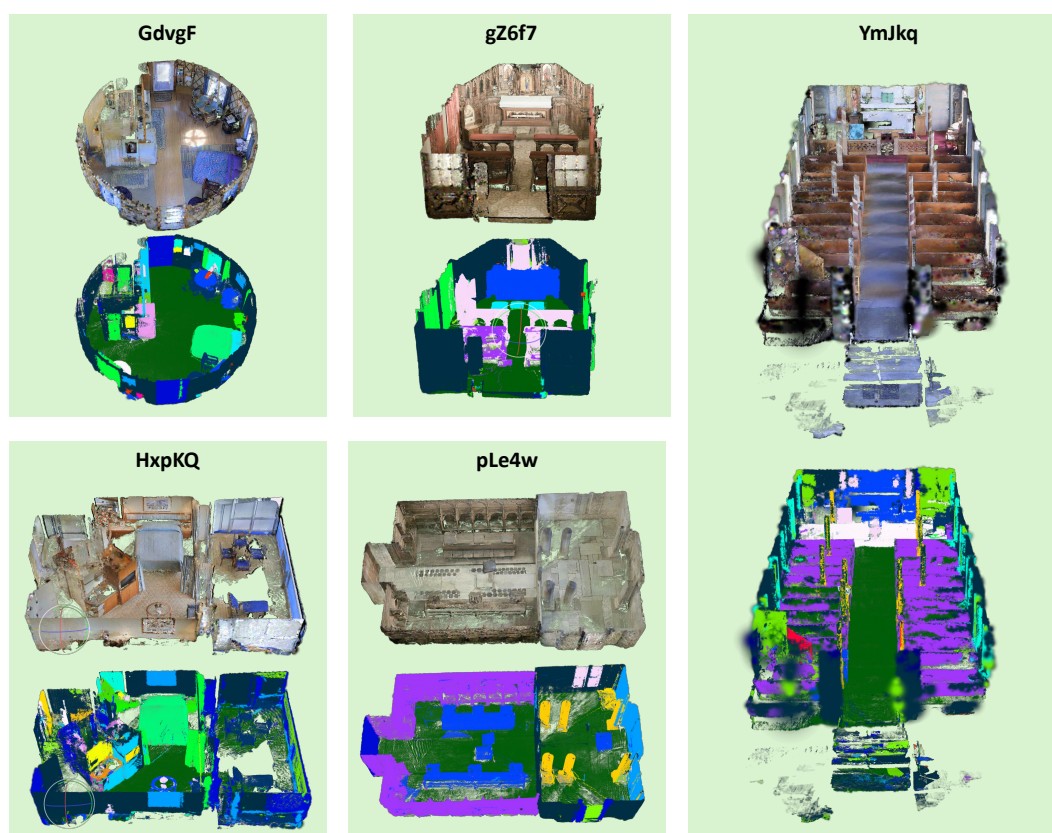

Figure S.4: **RGB and Semantic Reconstruction for MP3D.**

2. **Matterport3D Dataset** [103]
   - URL: `https://niessner.github.io/Matterport/`
   - License: Non-commercial (`https://kaldir.vc.in.tum.de/matterport/MP_TOS.pdf`)

**Software.** We use Habitat-Sim as our simulation environment and develop a custom sparse raster-ization CUDA toolkit based on 3D Gaussian Splatting. For mapping, we adopt SplaTAM as the backbone and fine-tune OneFormer to serve as our semantic camera. During evaluation, we also implement SGS-SLAM for comparative analysis. The source code for these components is available at:

1. **Habitat-Sim** [101]
   - URL: `https://github.com/facebookresearch/habitat-sim.git`
   - License: MIT

2. **3D Gaussian Splatting (3DGS)** [25]
   - URL: `https://github.com/graphdeco-inria/gaussian-splatting.git`
   - License: Custom (`https://github.com/graphdeco-inria/gaussian-splatting?tab=License-1-ov-file#readme`)

3. **SplaTAM** [51]
   - URL: `https://github.com/spla-tam/SplaTAM.git`
   - License: BSD-3-Clause

4. **SGS-SLAM** [7]
   - URL: `https://github.com/ShuhongLL/SGS-SLAM.git`
   - License: BSD-3-Clause

