# OpenReview forum: "Understanding while Exploring: Semantics-driven Active Mapping"
_NeurIPS.cc/2025/Conference — NeurIPS 2025 poster_

### Official Review · Reviewer_f7pY · 2025-06-30

**Clarity:** 3
**Significance:** 3
**Originality:** 3
**Rating:** 5
**Confidence:** 3

**Summary:**

The paper introduces ActiveSGM (Active Semantic Gaussian Mapping), a novel framework for active semantic mapping. This system combines semantic and geometric uncertainty quantification, guiding robots to explore their environments efficiently by strategically selecting the most informative viewpoints. It utilizes a 3D Gaussian Splatting (3DGS) backbone to generate dense, high-fidelity semantic maps, enhancing exploration and scene understanding. The framework has been tested on the Replica and Matterport3D datasets, showcasing superior performance in improving mapping completeness, accuracy, and robustness to noisy semantic data.

**Questions:**

See Weakness

**Ethical Concerns:**

["NO or VERY MINOR ethics concerns only"]

**Limitations:**

See Weakness

**Quality:**

3

**Strengths And Weaknesses:**

Strength
1. The combination of active exploration with semantic and geometric uncertainty provides a meaningful advancement in robotic mapping.
2. The paper demonstrates its effectiveness through comprehensive experiments on popular datasets like Replica and MP3D.
3.
Weakness
1. The use of pseudo labels from the pre-trained OneFormer model for semantic predictions introduces a dependency on external, potentially error-prone models. This could lead to inaccuracies, especially in complex or occluded scenes where the pretrained model might perform suboptimally.
2. The method may be limited by the initial semantic map’s accuracy, as semantic initialization from first observations can be unreliable, especially in occluded or unseen regions.

---

> ### Author Rebuttal · Authors · 2025-07-30
>
> We sincerely thank the reviewer for the feedback. We’re grateful for your recognition of the **novelty and value** of combining active exploration with semantic and geometric uncertainty, and we appreciate your acknowledgment of our **strong performance** across comprehensive experiments on the Replica and MP3D datasets.
>
> ```
> Q4.1 Dependency on semantic predictions
> ```
>
> Yes, relying on pseudo labels from a pre-trained OneFormer model can introduce errors, particularly in complex or occluded scenes. However, this reflects **a more realistic setting**, as ground-truth semantic masks are never available in real-world applications. ActiveSGM is explicitly designed to handle such noisy inputs.
> First, our use of **a soft semantic representation** allows the model to *retain uncertainty rather than commit to potentially incorrect labels*.
> Second, the **semantic exploration criterion** actively guides the agent to *revisit uncertain or ambiguous regions*, effectively reducing semantic noise through repeated observations.
> As shown in our experiments, this approach enables ActiveSGM to **refine the initial pseudo labels over time**, resulting in robust semantic reconstruction despite the inherent limitations of pre-trained segmentation models.

---

> ### Author Response · Authors · 2025-08-05
>
> Dear Reviewer,
>
> Thank you again for your detailed feedback and for taking the time to review our submission. We hope our rebuttal has addressed your concerns clearly. If you have any additional questions or thoughts, we would greatly appreciate the opportunity to clarify further during the discussion period.
>
> Please let us know if there is any aspect of the paper or our response that you’d like us to elaborate on.
>
> Best regards,
>
> Authors

---

### Official Review · Reviewer_MRuN · 2025-07-03

**Clarity:** 4
**Significance:** 3
**Originality:** 3
**Rating:** 5
**Confidence:** 5

**Summary:**

The authors present the first method for active semantic mapping with neural radiance fields. They propose a representation in which cropped semantic probability vectors are stored in a 3DGS backbone. The objective for active mapping simultaneously incentivizes regions of high semantic entropy computed from the semantic probability vectors and low geometric coverage computed from missing pixel counts. This approach is validated on the Replica and Matterport3D datasets and shows additional capability over alternate approaches while simultaneously achieving strong (often state of the art) performance on accurate and efficient mapping.

**Questions:**

I am confused about the notation around when and how the top k probability class values are cropped in the semantic vector. It seems to me that \tilde{P} is the cropped semantic vector and P is the uncropped version. However, \tilde{P} is only used once in the method section, but it seems that the vector would necessarily be cropped prior to entropy calculations during . Could you clarify?

Did you perform hyperparameter tuning to find the chosen values of each lambda in equation 5? Is the method very sensitive to the choice of hyperparameters?

Do you think that your method could be extended in future work to incorporate open vocabulary semantic segmentation models? Or would it require rethinking the entire approach?

**Ethical Concerns:**

["NO or VERY MINOR ethics concerns only"]

**Final Justification:**

This method fills a clear gap in current literature as the first method for active semantic mapping with neural radiance fields. The authors present a novel and clear approach. The literature review is thorough and educational across the different axes of research that come together to make the contribution in this work. The experiments are thorough and show a clear advantage over the baselines. The baselines are well selected and the differences between them and the new approach is clearly explained. The tables and figures are easy to understand.

**Limitations:**

Yes, particularly in the supplement.

**Quality:**

4

**Strengths And Weaknesses:**

Overall, this is a strong paper which I enjoyed reading. The method fills a clear gap in current literature and is a novel and well presented approach. The literature review is thorough and educational across the different axes of research that come together to make the contribution in this work. The experiments are thorough and show a clear advantage over the baselines. The baselines are well selected and the differences between them and the new approach is clearly explained. The tables and figures are easy to understand.

As a suggestion for improvement, the results in the supplement regarding runtime performance are significant and should be included in the main paper with extra space on publication. In addition, the limitation regarding localization and execution should be ultimately included in the main paper. The immediate next question regarding the impact and significance of this work is the ability to run it effectively on a physical robot in real time since no hardware experiments were provided. Evidence in this benchmarking is sufficient to convince me that it would be possible, and further discussion regarding the execution and localization limitations would help inform extensions of this work.

There is a missing parenthesis in equation 7.

---

> ### Author Rebuttal · Authors · 2025-07-30
>
> We sincerely thank the reviewer for the thoughtful and encouraging review. We truly appreciate your recognition of the **novelty and clarity of our method**, the **thoroughness of our literature review and experiments**, and the **clear explanation and design of our tables and figures**. We are also grateful that you found the paper **enjoyable to read** and acknowledged its **potential impact and relevance** to the research community.
> We respond to the suggestions and concerns as following:
>
> ```
> Q3.1 Semantic vetor and Entropy
> ```
> Each Gaussian stores only a cropped semantic vector for memory efficiency, but entropy is computed using the full vector during rendering (see supplement Line 20, Sparse Rendering). In practice, low-probability categories contribute negligibly to the entropy since $plog(p)$ approaches zero, so the approximation remains accurate.
>
> ```
> Q3.2 No real-world hardware experiments
> ```
> Deploying and evaluating based on physical robots is challenging due to the lack of well-annotated data, especially to get the grounth-truth semantic label for evaluation. For this reason, prior methods are evaluated only in simulation, making photorealistic environments the most practical and fair testbed for benchmarking. We will consider to extend our work into real-world application in the future.
>
> ```
> Q3.3 lambda values
> ```
> Yes, the λ values in Equation 5 were selected through empirical tuning, but the method is not highly sensitive to these choices. As long as the two semantic loss terms are balanced in terms of numerical magnitude, the pipeline remains stable and performs well.
>
> ```
> Q3.4 Support Open-Vocabulary Model
> ```
>
> We consider this an exciting direction for future work. Incorporating open-vocabulary semantic features would require reliable uncertainty quantification to ensure robust integration into our framework.

---

### Official Review · Reviewer_7urF · 2025-07-03

**Clarity:** 3
**Significance:** 3
**Originality:** 2
**Rating:** 4
**Confidence:** 4

**Summary:**

This paper proposes an active semantic mapping technique based on the ActiveGAMER active mapping technique.  The paper proposes a way to extend the Gaussian splatting map representation to include a sparse representation of the semantic information associated with a Gaussian component in the map, a way to update the semantic information using an uncertain measurement, as wel as incorporating the semantic information in the map into an exploration criterion used to determine the next best view used for active mapping.  The proposed approach is tested using a number of simulated scenarios, and evaluated against a number of different competing methods; it achieves good results that compares well with that of competing methods.

**Questions:**

- Are there any aspects of this review that are mischaracterisations of the proposed approach or the presentation thereof?

**Ethical Concerns:**

["NO or VERY MINOR ethics concerns only"]

**Final Justification:**

The authors' responses mostly confirmed my initial view of the paper:
- It is the first active semantic mapping technique based on the Gaussian splatting map representation.
- It is heavily based on the ActiveGAMER active mapping technique, which is simply extended to incorporate semantic information.
- It achieves good results, but the experiments could have been designed better to be more informative.

**Limitations:**

Yes

**Quality:**

2

**Strengths And Weaknesses:**

### Quality:
The proposed approach can be viewed as a fairly simple addition of semantic representation to an existing active mapping approach.  This is done fairly well and is supported by good results; however, this aspect of the paper has its weaknesses:
- One of the main contributions of the paper is the augmentation of a Gaussian splatting map representation with a distribution over semantic labels.  This distribution is approximated to reduce the storage complexity of the representation.  The approximation is a very simple one: The $k$ most likely labels as seen *from the initial observation* (line 191) are stored and updated.  This will most likely cause the proposed approach to be brittle, since it cannot recover from a case where the correct semantic label is incorrectly discarded based on the initial observation (it conceivable that this could regularly happen in certain types of scenarios).  An obvious alternative representation is to retain a distribution over all the semantic labels, but to group all the labels with a low likelihood into a "super category"; this would allow a similar reduction in storage requirement as that of the proposed approach, but would allow the mapping technique to recover from inaccurate semantic observations in the initial view.
- Another contribution is the incorporation of the semantic map representation into the exploration criterion used for active mapping.  However, this is done very simply (Eq. 8) and is unmotivated; this aspect of the paper could have benefitted from a more principled development.
- The proposed approach is a simple adaption of the active mapping technique ActiveGAMER (ref. 74 in paper; see "Originality" for a discussion).  As a result, the proposed approach inherits weaknesses inherent to ActiveGAMER.  For example, the proposed approach seems to perform active mapping by calculating the next-best view, and then navigating to this view (Fig. 1).  This is quite a rudimentary form of active mapping, and it would have been better to plan a trajectory based on the expected information gain *along the trajectory*.  It is also not clear how the proposed approach could be extended to function in real-world scenarios, where it might not be possible to reach the calculated next-best view, and even if it is reachable, the very simple motion cost model used would probably be inappropriate.

The proposed approach was tested in a number of different ways and evaluated against several other competing methods.  However, this aspect of the paper also has some minor weaknesses:
- The proposed approach is only tested in simulated scenarios.  Although the simulation environments are widely used and photorealistic, the lack of testing on real-world data causes uncertainty about the applicability of the proposed approach to real-world scenarios.  Testing active mapping in the real world would have been a significant undertaking though.
- The paper evaluates the proposed approach against a range of different techniques.  However, some of the competing techniques address a different problem (SLAM vs. mapping, and active *semantic* mapping vs. active mapping), and some of the competing techniques are not evaluated in exactly the same way (see discussions under "Clarity" below).  This makes it difficult to reliably draw conclusions from the results of the experiments.  The paper could have benefitted from a more systematic evaluation, isolating the causes for the differences in results seen by the evaluated methods. For example, since the semantic representation chosen for the proposed approach is one of the main contributions of the paper (see "Originality" below), an experiment that directly compares the proposed representation with alternatives would have provided a much better way to evaluate this design choice.

### Clarity:
The paper is mostly well written, clear and neat.  The ideas presented in this paper and how they are related are quickly clear on an overview level after a quick read-through.  However, when reading carefully through the paper, a number of issues with the presentation become clear, including the following:
- The reasons for reviewing SLAM techniques (lines 76-84 and 118-140) and evaluating against SLAM techniques (Sec. 4) are not clear, since the paper proposes a semantic *mapping* technique.  If the goal is to review and evaluate against the *map representation* of existing SLAM techniques, then it has to be done in a much more motivated and nuanced manner than is done in the paper, since the contexts are different.  The statement in line 84: "We use SplaTAM [51] as the SLAM backbone of our algorithm," which is obviously not accurate, compounds this issue.  For readers not already familiar with the field, it would be beneficial to mention the difference between mapping and SLAM.
- Although the related work is discussed extensively and clearly, the relationship between the proposed approach and the related work could have been presented better.  For example, two active semantic mapping techniques, SSMI and TARE (refs. 100 and 101 in the paper) are evaluated against, but they are not included in the literature review of Sec. 2, and they are therefore not characterised at all.
- The proposed method uses local and global keyframes (see Fig. 1 and lines 226-230), but their meanings and the distinction between them are never given (the discussion in lines 226-230 provides no help).
- The system overview in Fig. 1 shows a path planner that forms part of the exploration module; however, this path planner is not described anywhere in the paper.
- The Gaussian splatting representation used in the proposed approach, as well as the loss function used for training are discussed in Sec. 3.1, but the training procedure is not discussed at all.  For example, it is unclear how Gaussian components are generated and how their parameters are subsequently determined.
- There are several aspects of the map representation that is unclear or ambiguous, including:
  - According to the rendering procedure in Eq. 1, *every* Gaussian component in the map will infuence *every* pixel in the image.  There is presumably some cut-off where the effect of a Gaussian component below a certain threshold is excluded, but this is not mentioned at all.
  - The map representation seems to require RGB-D inputs (see lines 176 and 179), but this requirement is never explicitly stated, and it is not clear whether the proposed approach can be applied to RGB images.
  - The meaning of $f_i(\mathbf{p})$ is not given (line 177), and consequently Eq. 1 cannot be interpreted with certainty.
  - For the definition of the semantic loss (lines 204-219), the OneFormer observations are used as ground truth and the projected label distributions as predictions.  To remove uncertain "ground truth" labels, a mask is defined based on the entropy of the distribution over the labels (line 207), but this entropy is then defined in terms of the *predicted* label distribution (Eq. 4), which does not make sense to me.
- In the definition of the geometric exploration criterion (Eq. 6), the pixels that have a nonzero silhouette value are counted.  However, according to the definition of the silhouette in Eq. 1, no pixel will have a silhouette value that is exactly zero (except at initialisation), and the geometric exploration criterion in Eq. 6 therefore simply counts the number of pixels in the image (which does not make sense).  There is presumably some cut-off where the silhouette projection to a pixel that falls below a certain threshold is considered to be zero, but this is not discussed at all.  The motivation for the way in which the geometric and semantic exploration criteria are combined (Eq. 8) is also not given.
- The results initially appear to be quite good, but at closer inspection, there are some issues with the experimental setup and results that are not properly discussed, and which create some uncertainty.  These include:
  - It is unclear why the F-1 scores in Table 1 are only give for some entries.
  - It appears that results of the first 4 rows of Table 1 have been lifted from the SGS-SLAM paper (ref. 7 in the paper), but this is not mentioned, and there is therefore questions about whether the experimental setups are the same and consequently whether the comparison is fair.
  - The first 4 rows of Table 1 are all results of SLAM methods, which addresses a different (although related) problem than that of the proposed approach, which addesses the mapping problem.  An assumption of the mapping problem is that the camera poses are known with certainty, whereas for the SLAM problem, the poses are estimated.  The availability of exact knowledge of the camera poses could possibly be a significantly advantage for the proposed approach when comparing the quality of the maps; however, this fact is not discussed in the interpretation of the results.
  - Line 328 seems to say that results for the first 3 rows in the red band in Table 1 have been lifted from the paper by Zhang *et al.* (ref. 92 in the paper); however, it was not clear to me whether this was the case (I could not find the results quoted in Table 1 in the paper by Zhang *et al.*).  In addition, lines 326-328 state that the proposed approach is not evaluated on exactly the same data as the competing methods, and it is consequently uncertain whether any reliable conclusion can be made from a comparison between the results.
  - Fig. 3 illustrates issues with the predicted semantic labels for SGS-SLAM in one scene and compares it with the ground-truth labels.  However, the predicted labels for the proposed approach for the same scene is not shown, and the relevance of this figure is therefore not clear.  In addition, it is unclear whether the issues are influenced by the pose uncertainty inherent to the SLAM problem, which makes the relevance of the figure even less clear.
  - Line 294 states that top-1 and top-3 accuracy results are given, but I could not find it anywhere in the paper.

### Significance:
Despite the limited nature of the new ideas presented in this paper (see "Originality" below), the good results obtained in the experiments and simplicity of the new ideas give an indication that the ideas will most probably be adopted by the research community.

### Originality:
The technique proposed in this paper is heavily based on the active mapping technique ActiveGAMER (ref. 74 in the paper).  Specifically, the geometric and colour map representation and approach to learning the map, the exploration map and approach to select the next-best view, the path planner, and the keyframe scheme are identical.  Essentially, the proposed approach only adds:
- a representation for incorporating semantic information into the map,
- a way to project the semantic information stored in the map to an image and use it to train the map, and
- the incorporation of the semantic information in the map into an exploration criterion.

Nevertheless, the proposed does indeed seem to be the first active semantic mapping technique based on the Gaussian splatting map representation (lines 349-350).  Good results also support the contribution.

A note about contextualisation: Although ActiveGAMER is appropriately mentioned and referenced, the extend to which the proposed approach is similar to ActiveGAMER is only clear after a careful read of the paper.  A clearer contrast between the two methods would have been better.

### Other:
There are a number of minor issues in the paper, including the following:
- Line 82: "TSDF" is undefined.
- Lines 124, 126 and 141: Quotation marks should be formatted better.
- Line 141 states that "All the above approaches are "passive"", but this is not true since active mapping is discussed in lines 85-105.
- Line 176: $c_i$ should be $\mathbf{c}_i$, and $d_i$ is undefined.
- Eq. 7: A bracket is missing.
- Table 1: The meaning of "GT" is undefined; the reader has to guess that this refers to the ground truth.
- Lines 320-321: "and introduce arbitrary" -- sentence incomplete

---

> ### Author Rebuttal · Authors · 2025-07-30
>
> We sincerely thank the reviewer for recognizing the strengths of our work. We appreciate the positive remarks on the **clarity and simplicity of the method**, as well as the acknowledgment of its **strong empirical performance**. We're especially grateful for the recognition that the approach shows **potential for impact and adoption by the research community**.
> Here, we respond to the suggestions and concerns raised by the reviewer.
>
>
> ```
> Q2.1 [Quality] "Super category" for semantic representation
> ```
>
> Thanks for the suggestion. We will explore integrating "super categories" for other classes in future work.
>
> ```
> Q2.2 [Quality] Motivation of the exploration criterion
> ```
>
> Our semantic exploration criterion is designed to favor observations that
>
> (1) cover previously unseen regions (areas without existing Gaussians),
>
> (2) reduce semantic uncertainty (entropy of the predicted class distribution), and
>
> (3) low travel cost to encourage efficient exploration.
>
> These components are combined to form a practical and lightweight criterion that balances coverage, semantic gain, and efficiency. A similar heuristic is used in Ref [71], Eq. 9.
>
> ```
> Q2.3 [Quality] Next-best-view vs. entire trajectory planning
> ```
> We use step-wise next-best-view planning with a shorter horizon (0.5 m) to enable more efficient and adaptive exploration. Long-horizon trajectory planning, like in He et al. (mentioned by Reviewer fEfZ), requires dense sampling and costly post-optimization. In practice, planning is cheaper than moving the robot. Our method integrates new observations on-the-fly and avoids continual learning issues in NeRF.
>
> ```
> Q2.4 [Quality] Real-World Applicability
> ```
> We account for real-world constraints by maintaining an occupancy grid with a 0.2 m collision buffer, restricting the agent to move only within free voxels. These ensure that planned viewpoints are both reachable and obstacle-free.
>
> ```
> Q2.5 [Quality] Simulation vs. Real World
> ```
> Real-world evaluation is challenging due to the lack of well-annotated RGB-D-semantic datasets and the difficulty of deploying all baselines on physical robots. For this reason, prior semantic mapping methods are also evaluated only in simulation, making photorealistic environments the most practical and fair testbed for benchmarking.
>
>
> ```
> Q2.6 [Quality] Comparisons with Competing Methods
> ```
>
> We thank the reviewer for highlighting the concern regarding comparing our method with a range of techniques addressing slightly different problems. We agree that such comparisons need to be handled carefully, and we clarify our intentions and methodology as follows:
>
> **Semantic-SLAM vs. Semantic Mapping:**
>     To the best of our knowledge, most NeRF/3DGS-based semantic mapping methods—such as NIDS-SLAM, DNS-SLAM, SNI-SLAM, and SGS-SLAM are formulated as semantic SLAM systems that jointly solve for localization and semantic mapping. In contrast, our method assumes known camera poses, focuses solely on semantic mapping. Therefore, we agree that direct performance comparisons may not be entirely fair, as SLAM inherently involves the harder task of localization.
>     Furthermore, many SLAM methods use ground truth semantic labels, while our method uses noisy semantic predictions from a pre-trained segmentation model, adding further difficulty to our task.
>     To account for this, we include the yellow band results in Table 1 solely for reference and completeness. We explicitly do not claim superiority over SLAM systems in this context.
>
> **Fair Comparison via Controlled Setup (Table 1 – SGS-SLAM variant):**
>
> To enable a fairer comparison with semantic SLAM methods, we select SGS-SLAM as the strongest representative baseline and reconfigure it to function as a semantic mapping-only system:
> (1) Disable its tracking module and use ground truth poses.
> (2) Replace ground truth semantics with semantic predictions.
> This allows us to directly evaluate the semantic representation quality, which is a key contribution of our method. We then disable our own active exploration module to ensure a passive, mapping-only setting, enabling a fair apples-to-apples comparison between semantic representations.
>
> **Active vs. Passive Mapping (Table 1 – Blue):**
> By comparing ActiveSGM (with exploration enabled) against its passive variant and against the above baselines, we highlight our method achieves better segmentation with fewer observations/steps, demonstrating the effectiveness of our active exploration strategy guided by semantic and geometric uncertainty.
>
> **Active 3D Mapping and View Synthesis (Tables S.6, S.7):**
> While some baselines focus on active mapping tasks without semantics (e.g., for 3D reconstruction or novel view synthesis), we ensure fair comparisons by using posed RGB-D input across methods. Our semantics are predicted from RGB data, not an extra input. This allows us to show that even with semantic reasoning added, our system remains competitive in standard active mapping tasks.
>
> **Active Semantic Mapping Baselines (Table 1 – Red):**
> We also include comparisons to recent active semantic mapping baselines, which are more aligned with our task definition. These further validate the advantage of our sparse semantic representation and active policy.
>
> We hope this clarifies the rationale behind our evaluation setup and demonstrates that we took significant care to ensure fair and interpretable comparisons across different problem settings. We will make these clarifications more explicit in the final version of the paper.
>
> ```
> Q2.7 [Quality] Evaluation of Semantic Representation
> ```
>
> Please also see supplement Table S.4, in which we also train ActiveSGM in a passive maner using the same frames of SGS-SLAM. The result shows that our sparse semantic representation maintains more accurate semantic understanding than RGB-encoding representation.
>
> ```
> Q2.8 [Clarity] SLAM or Mapping
> ```
> We will clarify the related work. Here we mainly focus on the mapping moudle of previous methods, specificlly for those using 3DGS or NeRF as mapping backbones.
>
> ```
> Q2.9 [Clarity] SplaTAM Misstatement
> ```
> Apologies for the misstatement — we use the mapping backbone of SplaTAM, not the full SLAM system. We will correct this in the revised version.
>
> ```
> Q2.10 [Clarity] Missing related work discussion (SSMI and TARE)
> ```
> These were missed due to oversight and will be included in the final version. They are used as baselines in Ref [92].
>
> ```
> Q2.11 [Clarity] Keyframe Definitions
> ```
>
> Please see L220-230. Local keyframes are the those have the highest overlap within all observations, as designed in SplaTAM. The global keyframes are described in L226-230, and are designed for maintaining global semantic information.
>
> ```
> Q2.12 [Clarity] Path Planner
> ```
>
> Due to the limited pages, we described the path planner in the supplement, L 51-58. The complete path planner is: we use the exploration criterion to decide the destination, and use the local path planner to decide the path from the current position to the destination.
>
> ```
> Q2.13 [Clarity] Questions about 3DGS
> ```
> We follow the standard training scheme for the Gaussian components as in SplaTAM (Ref [51]) and 3DGS (Ref [25]); please see Supplement L146–L150 (all training configurations are available online). Equation 1 follows the standard volumetric rendering process using alpha blending. $f_i(p)$ in Eq. 1 denotes the probability that a ray stops at sampling position $p$ (see also Eq. 3 in Ref [22]). In practice, we apply an early stopping criterion when the accumulated transmittance drops below 0.01, effectively ignoring negligible Gaussian contributions.
>
> ```
> Q2.14 [Clarity] RGB-D Requirement
> ```
> Our method requires RGB-D input, as we focus on semantic mapping with accurate geometry. If only RGB images are available, an additional depth estimation module would be needed to generate the necessary 3D structure.
>
> ```
> Q2.15 [Clarity] Entropy-Based Filtered Mask
> ```
> Apologies for the confusion. The entropy filter at L207 is applied on the OneFormer prediction, not on the rendered semantic mask.
>
> ```
> Q2.16 [Clarity] Geometric Exploration Criterion
> ```
> The unobserved regions do not contain any Gaussians, so the silhouette mask value in those pixels is 0 by definition.
>
> ```
> Q2.17 [Clarity] F-1 / Top-1 / Top-3 Accuracy
> ```
> Other baseline methods report only mIoU. To provide a more comprehensive evaluation, we select the strongest baseline (SGS-SLAM) and additionally compute F-1, mAP, Top-1, and Top-3 accuracy, as shown in Table S.4 of the supplement.
>
> ```
> Q2.18 [Clarity] Yellow Results in Table 1
> ```
> Yes, the yellow-band results in Table 1 were copied from the original publications. We include them for reference because they are among the few methods that use 3DGS or NeRF as a semantic representation.
>
> ```
> Q2.19 [Clarity] SLAM vs. Mapping Results
> ```
> For the experiments in Table 1 (blue) and Tables S.4–S.7, all baselines are run with RGB-D inputs. To ensure fairness, we disable the tracking modules in SLAM baselines and use ground-truth camera poses.
>
> ```
> Q2.20 [Clarity] Results from Zhang et al.
> ```
> Due to space constraints in the main paper, we report averaged scores from Table 2 of Zhang et al. Full per-class results are provided in Table S.5 of the supplement.
>
> ```
> Q2.21 [Clarity] Purpose of Figure 3
> ```
> The purpose of Figure 3 is to illustrate the color-coding ambiguity in RGB-encoded semantics used by SGS-SLAM. In contrast, ActiveSGM maintains a global semantic distribution, avoiding such ambiguity. To eliminate pose-related confounds, we provide ground-truth poses to SGS-SLAM in this comparison.
>
> ```
> Q2.22 [Originality] Significance of ActiveSGM, v.s. ActiveGAMER
> ```
> Please refer to our comparison between ActiveGAMER and ActiveSGM (Q1.2), where we discuss the significance of our contributions in terms of accuracy, efficiency, and semantic fidelity.
>
>
> ```
> Q2.23 Other minor issues
> ```
> Thanks for pointing out the issues. We promise to fix them in the final version.

---

> > ### Comment · Reviewer_7urF · 2025-08-03
> >
> > Thank you to the authors for their response.  I have carefully read all the reviews and the authors' responses.
> >
> > I appreciate the authors' efforts to address to the points raised in my review -- many of the aspects that were unclear to me have now been clarified.  My initial assessment of the paper has been mostly confirmed, and it can be characterised as follows:
> > - It is the first active semantic mapping technique based on the Gaussian splatting map representation.
> > - It is heavily based on the ActiveGAMER active mapping technique, which is simply extended to incorporate semantic information.
> > - It achieves good results, but the experiments could have been designed better: the mapping component of SGS-SLAM is extensively used as comparison despite the fact that its representation is chosen for *localisation and* mapping, and much of the comparative results have been lifted from papers and are not direct comparisons (Table 1).  I want to echo the sentiment of Reviewer fEfZ that the results and comparisons presented do not provide much insight into the reasons for the performance of ActiveSGM.
> >
> > In conclusion, my view of the paper has not change significantly, and I have consequently decided to keep my score unchanged.

---

> > > ### Author Response · Authors · 2025-08-05
> > > **Response to the comment**
> > >
> > > We thank the reviewer for carefully reassessing our response and for acknowledging the clarified points. We appreciate your recognition of our work as the **first active semantic mapping method based on 3D Gaussian Splatting** and its overall strength.
> > >
> > > ```
> > > Detailed comparison with ActiveGAMER
> > > ```
> > >
> > > Due to the rebuttal word limit, we could not fully explain the differences between ActiveGAMER and ActiveSGM in **Q1.2**, so we elaborate here. We respectfully disagree with the characterization of our method as a simple extension of ActiveGAMER. While we reuse some infrastructure (e.g., local planner, exploration space), **ActiveSGM introduces fundamental changes** to both the **mapping representation** and **exploration policy**, including:
> > >
> > > - **Sparse probabilistic semantic representation** stored in the 3D Gaussians, maintaining a distribution over the most likely classes instead of projecting to RGB. This enables efficient, expressive semantic reasoning with minimal memory overhead.
> > > - A custom **sparse semantic rendering module** that allows efficient projection directly from the sparse semantic Gaussian representation without dense label processing.
> > > - A new **semantic-aware exploration criterion** that complements geometric uncertainty. This allows the system to prioritize informative observations that improve completeness and semantic accuracy.
> > > - Unlike ActiveGAMER, which passively benefits from occasional re-observations, ActiveSGM intentionally revisits previously explored regions where semantic predictions are uncertain, leading to targeted refinement.
> > > - Thanks to this **on-the-fly revisiting and refinement design**, our system achieves high-quality semantic mapping during exploration itself, **eliminating the need for post-hoc photometric refinement**.
> > >
> > > Together, these contributions form a complete and significant extension beyond ActiveGAMER.
> > >
> > >
> > > | Component                      | ActiveGAMER                                     | ActiveSGM (Ours)                                                                  |
> > > |-------------------------------|-------------------------------------------------|----------------------------------------------------------------------------------|
> > > | Base Representation           | 3DGS (geometry + color)                         | 3DGS (geometry + color + **sparse probabilistic semantic representation**)                         |
> > > | Semantic Representation              | N/A                                             | **Per-Gaussian class distributions**                                             |
> > > | Semantic Rendering            | N/A                                             | **Sparse semantic renderer**                                      |
> > > | Keyframe Selection       | 3D overlapping + PSNR                      |     3D overlapping + **Uncertainty of pseudo semantics**                                         |
> > > | Exploration Criterion         | Geometric completeness                      | Geometric completeness + **Semantic uncertainty**                                             |
> > > | Revisiting Behavior           | Incidental re-observations                      | **Intentional revisiting of uncertain semantic regions**                         |
> > > | Post-Refinement               | Required (photometric)                         | **Not needed**                                                                   |
> > >
> > >
> > > We hope this response clarifies the conceptual and technical distinctions between ActiveSGM and ActiveGAMER, and we appreciate the opportunity to further explain the significance of our contributions.
> > >
> > >
> > > ```
> > > Experiement design and comparison
> > > ```
> > > We thank the reviewer for raising this important concern. We agree that **running baseline methods under a unified experimental setup** is ideal for fair comparison. However, many of the prior works referenced in Table 1 are **not open-sourced**, and due to their system complexity, reimplementation may result in incorrect or inconsistent reproduction of their reported performance. For this reason, we opted to **quote their original numbers directly**, as done in prior evaluations.
> > >
> > > For fair comparison, we focus on **SGS-SLAM**, which is open-sourced and shares the SplaTAM-based mapping backbone. We **disabled its tracking module** and used **ground-truth poses** to isolate the mapping component. Additionally, we replaced ground-truth semantics in SGS-SLAM with **predicted labels from OneFormer**, aligning it with our setup.
> > >
> > >
> > > Regarding the scope of direct comparisons: to our best knowledge, and as acknowledged by the reviewer, **ActiveSGM is the first method to perform active semantic mapping with 3D Gaussian Splatting**, so no prior work exists for direct comparison. In Table 1, we have included comparisons to existing **active semantic mapping** works based on other representations, to help contextualize our results.
> > >
> > > We will clarify these points more explicitly in the final version.

---

> > > > ### Comment · Reviewer_7urF · 2025-08-07
> > > >
> > > > Thank you for the clarification about the comparison with ActiveGAMER as well as the motivation behind the experimental setup and presentation of results.  Since the authors disagree with my characterisation of the differences between ActiveGAMER and ActiveSGM, I want to make sure that I understand the nature and magnitude of all the differences that they highlighted.  To this end, I list the differences and where I could see that they are described in the submitted manuscript below:
> > > > - **Sparse semantic representation**: lines 190-195
> > > > - **Per-Gaussian class distributions**: also lines 190-195
> > > > - **Sparse semantic renderer**: lines 196-203, including Eq. 3
> > > > - **Uncertainty of pseudo semantics for keyframe selection**: lines 226-230, using Eq. 4
> > > > - **Semantic uncertainty used in exploration criterion**: Eq. 7, which uses Eq. 4, and which is incorporated in the overall criterion in Eq. 8
> > > > - **Intentional revisiting of uncertain semantic regions**: I could not immediately see where this is described.  Is this a characterisation of the behaviour resulting from the chosen exploration criterion, or did I miss an additional mechanism?
> > > > - **Post-refinement**: This component of ActiveGAMER was simply removed, right?
> > > >
> > > > Could the authors please comment on whether this list is accurate?
> > > >
> > > > Apart from the theoretical description of ActiveSGM presented in the manuscript, there could of course also have been practical challenges in extending ActiveGAMER that might not be clear from the manuscript.  Do the authors want to comment on any such practical challenges that could contribute to the assessment of the contribution?

---

> > > > > ### Author Response · Authors · 2025-08-07
> > > > > **Response to the comment 2**
> > > > >
> > > > > We sincerely thank the reviewer for carefully reading our paper and for the thoughtful and constructive feedback. Yes, your list for the components of our pipeline is correct. Following up on our discussion, we address the remaining questions below.
> > > > > ```
> > > > > Intentional revisiting of uncertain semantic regions
> > > > > ```
> > > > > Yes, this behavior emerges naturally from our exploration criterion. In ActiveSGM, we consider not only "unseen regions" (identified by the silhouette mask) but also the accuracy of semantic reconstruction. In practice, newly observed regions may reveal objects from ambiguous or challenging viewpoints, making classification difficult. For instance, in Replica Room 0, a top-down view of the blue round chair may resemble a round table. In such cases, the predicted semantic uncertainty (measured by entropy) encourages the agent to revisit the object from a different perspective to improve classification confidence.
> > > > >
> > > > > ```
> > > > > Practical challenges
> > > > > ```
> > > > > The biggest challenge in building an efficient active semantic mapping system lies in the high dimensionality of semantic features. We adopt the efficient rendering framework introduced by ActiveGAMER as our backbone. However, several issues remain:
> > > > > (1) It is infeasible to store high-dimensional features in every Gaussian due to memory constraints;
> > > > > (2) Even if we manage to fit these large features into GPU memory, rendering them is computationally expensive and undermines real-time performance (see Supplement L94–95);
> > > > > (3) Existing compression mechanisms, such as RGB encoding, often lead to ambiguous semantic representations.
> > > > > To address these challenges, we propose a novel top-k representation for semantic Gaussian splatting. However, this raises a second problem: if each Gaussian stores a different set of categories, how can we efficiently reconstruct the final dense semantic distribution? To solve this, we implement a custom CUDA module for sparse semantic rendering, which will be released upon acceptance.

---

### Official Review · Reviewer_fEfZ · 2025-07-14

**Clarity:** 2
**Significance:** 3
**Originality:** 2
**Rating:** 3
**Confidence:** 4

**Summary:**

The authors present an active mapping system based on 3D Gaussian splatting called Active Semantic Gaussian Mapping (ActiveSGM).
ActiveSGM represents the scene using 3D Gaussians (as opposed to NERFs) which is a new contribution. ActiveSGM infers the semantics of the visible surfaces and combines a geometric and semantic uncertainty criterion for planning exploratory actions to visit frontiers. ActiveSGM operates under a closed vocabulary assumption and shows robustness to noisy semantic predictions by leveraging predictions from a pre-trained model (relevant in real world systems) and balances memory constraints when dealing with many semantic classes in a scene by using the top-k semantic classes instead of all of them.
Another key part of the paper is using 3D Gaussians as a representation of the scene instead of NERFs. This is relatively new and the addition of semantics is a novel contribution to such systems.

**Questions:**

4.2 ReplicaSLAM
What part of the SGS protocol are we talking about here? Is there an active mapping component in this experiment?
If we’re just considering a pre-defined trajectory, then only point 1 (using OneFormer instead of ground truth) seems to be different from SGS-SLAM. The other points are only related to training evaluation and training data. In that case, what component of ActiveSGM are the authors trying to highlight with this experiment?
If we are considering an active mapping, then how are the number of steps coming into play - because SLAM systems would work with a pre-defined trajectory. So I am unable to understand the purpose of “steps”
What does the training refer to here? Does it refer to fine-tuning the OneFormer model? Based on my understanding, all the components mentioned in section 3 talk about the deployment phase of ActiveSGM, so what does training on 1/3 rd of the data mean for ActiveSGM. Additionally, while giving ActiveSGM a more difficult evaluation in terms of less training data and evaluation on novel views seems interesting, not providing the results while considering the same amount of data raises questions on the ceiling of the ActiveSGM performance - how much better is it compared to the baselines if given all the training data and evaluated on training scenarios or does the performance saturate. And what are the benefits compared to the baselines when the baselines are evaluated on the novel views and ActiveSGM is evaluated in novel views. The current set of of experiments shows different conditions for baselines and ActiveSGM and warrants more controlled evaluation conditions - the current experiment reads like the following - “We gave ActiveSGM less training data and also evaluated on novel views and it performs similar to some baselines and worse than the best” - it is important to have the apples-to-apples in the testing conditions to properly appreciate the system. Alternatively, if there are some holes in my understanding of the experimental setup then please point it out.
4.2 Replica (Novel Views)
ActiveSGM without exploration performs better than SGS-SLAM, thereby showing the benefits of the sparse representation.
It is hard to appreciate the benefits of the sparse representation because the paragraph does not properly emphasize the differences between ActiveSGM and SGS-SLAM. Is the following elaboration appropriate - “ActiveSGM without exploration is the same as using a predefined trajectory and doing SLAM. This is very similar to SGS-SLAM but SGS-SLAM compresses semantic information to r,g,b while ActiveSGM uses the sparse representation. ActiveSGM outperforms SGS-SLAM and hence shows the benefits of the sparse representation over compressing semantics into r, g, b?”. If so, then making this more clear will improve the clarity of the paper. If not, please explain how the authors arrive at that conclusion.
4.2 MP3D
The authors mention that compared to the baselines, ActiveSGM does not rely on ground truth semantic labels and still performs better. Based on the content in the paragraph, it is hard to understand what choices of ActiveSGM compared to the baselines lead to the improved performance. Is it the 3D Gaussian representation? Is it the exploration criterion?
While the individual components of ActiveSGM are explained in Section 3, not emphasizing the differences between ActiveSGM and the baselines in the experiments section makes it hard to understand “why” or “what components” might be contributing to better performance (especially for reader who aren’t extremely well versed with all the baselines)
It can be hard to provide an exact answer but a proper discussion on what changes and how that affected the results would provide clarity to readers.
MP3D experiment (NOT CRITICAL FOR REBUTTAL) - Did the authors consider using ground truth labels for ActiveSGM - would the performance get better or

**Ethical Concerns:**

["NO or VERY MINOR ethics concerns only"]

**Final Justification:**

I have read the rebuttal and the comments with the other reviewers. Overall, while I appreciate that there are differences in the proposed to prior work - in particular, ActiveGAMER - those are not particularly large deviations from current SOTA. The results also show relatively minor performance improvement. Furthermore, the work is limited to simulated datasets, though I appreciate the difficulty of running real world experiments. In sum, I do not feel that for this top venue, the work, at this time, makes a sufficient contribution.

**Limitations:**

See above.

**Quality:**

3

**Strengths And Weaknesses:**

Strengths
The paper describes a method for active dense mapping using 3D Gaussians as the representation. In addition, ActiveSGM is demonstrated with model predictions (without assuming ground truth semantic labels) and hence describes a good proof-of-concept for utilizing 3D Gaussians and semantic information for active mapping
The details of ActiveSGM are easy to follow
Weaknesses
My main criticism is related to the writing in the experiments section of the paper - as a reader I found it hard to grasp what components of ActiveSGM led to better performance over baselines in the individual experiments. I will elaborate on these in the Questions section.

Quality
The paper explains the components of the ActiveSGM pipeline well and is easy to follow.
The explanation of the exploration heuristics is also fairly intuitive and well explained.
The evaluation metrics and qualitative examples highlight capabilities of the proposed method
Clarity
The description of the ActiveSGM algorithm is clear. However, when I read the paper, it seems like I could not fully grasp the similarities and differences to previous methods which led to the improvements - a way to mitigate it might be to explicitly talk about the similarities and differences in the components (semantic representation, 3DGS vs NERF, ground truth vs semantic predictions, etc) while explaining the results from the experiments. In some cases the information is there but currently does seem to be emphasized enough. For example, the section on 3D Reconstruction and Novel View Synthesis shows how AvticeSGM is better than ActiveGAMER. But through the paper, I only understood the similarities to ActiveGAMER in terms of objectives and the fact that ActiveSGM does not have photometric refinement stage. So what are some reasons ActiveSGM performed better than ActiveGAMER?
Significance
The paper presents a method for leveraging 3D Gaussians as the scene representation for active mapping. ActiveSGM also incorporates a sparse representation of semantics and demonstrates active mapping with semantic predictions instead of utilizing ground truth semantic labels.
Originality
The method is original.
This paper on using NERFs for active perception seems relevant to the problem at hand (Active Perception using Neural Radiance Fields https://arxiv.org/abs/2310.09892). They try to maximize a predictive information objective. While they assume ground truth semantic labels, this paper could be an important related work. It would be nice to understand how this fits into the current work and an explanation of the benefits / drawbacks of this compared to ActiveSGM would be helpful to provide more clarity in the paper

---

> ### Author Rebuttal · Authors · 2025-07-30
>
> We thank the reviewer for the thoughtful review and encouraging feedback. We appreciate your recognition of our **practical setup** using predicted semantics, the **potential of combining semantic understanding with 3DGS**, and the **clarity of our pipeline and exploration heuristics**. We're glad the **evaluation effectively highlighted our method’s capabilities**.
> Here, we respond to the concerns and suggestionsraised by the reviewer.
>
> ```
> Q1.1 Experiments section lacks clarity on what drives performance
> ```
>
> We will substantially rewrite the experimental results section to better delineate concepts we inherited, such as benchmarks, metrics and baselines, and the contributions of our work.
>
> ```
> Q1.2 Advantage over ActiveGAMER is not clearly justified
> ```
>
> While ActiveGAMER focuses on actively acquiring geometric and photometric information for 3D reconstruction, ActiveSGM is specifically designed for semantic-aware active mapping. There are three key differences that contribute to ActiveSGM’s improved performance:
>
> 1. **Semantic-Aware Exploration**: ActiveSGM maintains a semantic map with sparse representation during exploration, enabling the system to reason explicitly about semantic coverage and uncertainty—something ActiveGAMER does not address.
>
> 2. **Joint Semantic and Geometric Criteria**: Our exploration strategy leverages both geometric completeness and semantic uncertainty to select informative viewpoints. This leads to more targeted exploration, particularly in semantically rich or ambiguous regions.
>
> 3. **No Post-Refinement Required**: Unlike ActiveGAMER, which relies on a post-exploration photometric refinement stage, ActiveSGM’s semantic feedback loop encourages natural revisiting of uncertain areas, making the mapping process more efficient and integrated.
>
> Together, these design choices allow ActiveSGM to produce higher-quality reconstructions with stronger semantic fidelity, even without relying on additional refinement stages.
>
> ```
> Q1.3 Component-wise differences to prior work are underexplained
> ```
>
> To address this, we highlight that our three core experiments are designed to isolate and evaluate specific aspects of our semantic mapping pipeline. The bullets below refer to Table 1.
>
> 1. **Replica-SLAM (yellow)**: We compare against NeRF/3DGS-based methods using dense RGB-encoded semantics and ground-truth labels, while our method uses only predicted semantics and still shows robustness by correcting noise over time.
> 2. **Replica-NVS (blue)**: We compare fairly against SGS-SLAM by disabling its tracking module and providing GT poses, isolating mapping performance and showing that our sparse semantic representation is both more memory-efficient and more accurate than dense alternatives.
> 3. **MP3D (red)**: We evaluate generalization in large-scale scenes and show that ActiveSGM, with full 3D path planning and no reliance on GT semantics, outperforms both geometry-only and semantic mapping baselines under the same RGB-D input.
>
> These comparisons collectively demonstrate the contribution of each design choice in our pipeline. We also refer the reviewer to the response Q2.6 to Reviewer 7urF for a more comprehensive comparison.
>
> ```
> Q1.4 He et al. on active perception is missing
> ```
>
> We will cite this paper. Compared to their method:
> - They require GT semantic masks; we use pseudo labels,which is more practical in real-world settings.
> - Their planning involves long-horizon, dense sampling; ours uses adaptive step-wise planning.
> - They retrain NeRF after each planning iteration; we optimize keyframes online without storing the full history.
> - They report no semantic metrics, limiting direct comparison.
>
> In contrast, ActiveSGM emphasizes semantic-guided, memory-efficient, online refinement.
>
>
> ```
> Q1.5 Clarification on the Active Mapping task
> ```
>
> Unlike passive mapping or SLAM, active mapping introduces **autonomous decision-making** into the mapping process. SLAM and passive mapping systems may operate on pre-recorded trajectories or under externally controlled by a human operator or predefined script. These systems passively consume data without influencing where the sensor moves or what information is acquired.
> In contrast, active mapping empowers the agent to plan its own trajectory in real time, actively selecting the most informative viewpoints and deciding when to stop exploration. The motivation behind active mapping is to achieve more efficient and targeted scene understanding by coupling perception with action, rather than relying on exhaustive or unguided data collection.
>
>
> ```
> Q1.6 [ReplicaSLAM]: Clarification on comparison protocol
> ```
>
> We appreciate the reviewer’s detailed questions and would like to clarify the setup and intent behind the ReplicaSLAM experiment (Yellow Band in Table 1).
>
> **Baselines**: The results are directly copied from SGS-SLAM [7]. These baselines operate under passive SLAM settings, taking RGB-D inputs with ground-truth semantic labels, and evaluate performance on the observed (training) frames. They rely on manually captured sequences and do not include any active exploration components.
>
> **Our Setup**: In contrast, our method operates under an active mapping setting, where the system autonomously selects which views to observe. We use posed RGB-D as input, and semantics are predicted on-the-fly using a finetuned OneFormer model. As our system plans its own trajectory, it does not observe the same frames used by the baselines—those frames are considered novel views to ActiveSGM.
>
> **Evaluation Protocol**: To ensure fairness and use the same evaluation testbed, we evaluate ActiveSGM on the same frames used by the baselines, even though our system has not seen these views during exploration. This allows a direct comparison in reconstruction quality while keeping the test set consistent.
>
> **Purpose of This Experiment**: This experiment is designed to highlight the effectiveness of our semantic mapping pipeline, particularly:
>
> (1) Our sparse semantic representation achieves accurate reconstruction even without access to ground-truth semantic labels.
>
> (2) Our semantic-guided exploration criterion helps select informative views, which improves both geometric and semantic reconstruction—despite relying on noisy pseudo labels.
>
> We hope this clarifies that the goal of the ReplicaSLAM experiment is not to compare exploration strategies directly, but rather to showcase the strength of our semantic representation and active observation selection under a more realistic, challenging setting.
>
> ```
> Q1.7 [ReplicaSLAM]: Clarification on "step", "training", and "data usage"
> ```
>
> We appreciate the reviewer’s questions regarding the meaning of “steps” and “training” in our experimental setup. The term “steps” is used in two contexts:
>
> 1.  For OneFormer, it refers to the number of fine-tuning iterations.
>
> 2. In SLAM and active mapping pipelines, “steps” refer to the number of observation frames used for optimizing the map representation. Typically, each observation frame results in one optimization step—thus, the number of steps is approximately equal to the number of observed frames.
>
> The term “training” also has two meanings:
>
> 1. For OneFormer, it refers to the offline fine-tuning process before deployment.
>
> 2. For SLAM/mapping, it refers to optimizing the scene representation from observations, which is commonly called “training” in radiance field literature.
>
> When we mention “training on 1/3 of the data,” we mean that ActiveSGM only observes and optimizes its scene representation using one-third the number of RGB-D frames, which may not necessarily correspond to the same observations used by the baselines. We will revise the text to clarify the terminology.
>
>
> ```
> Q1.8 [ReplicaSLAM]: Performance of ActiveSGM Trained on All Views vs. Active Pipeline
> ```
>
> Due to the limitation of pages, we cannot show all the experiments in the main paper. Please also see experiments in the supplement Table S.4, in which we also train ActiveSGM in a passive maner. As clarified in Q1.6, SGS-SLAM is trained and evaluated on the same trajectory. For a fair comparison, we construct a novel trajectory and evaluate both ActiveSGM and SGS-SLAM on it.
> The results show that even training on all views of the SLAM trajectory yields lower performance compared to our active pipeline. This supports the core motivation of ActiveSGM — actively selecting informative trajectories leads to better reconstruction and semantic understanding than passively using all available observations.
>
> ```
> Q1.9 [Replica (Novel Views)]: Experimental result interpretation (Sparse Semantic Representation)
> ```
>
> Yes, the interpretation is mostly correct. ActiveSGM without exploration is the same as using a predefined trajectory and doing *Semantic* SLAM *(w/o tracking)*.
>
> As we shown in Figure 3, the color-coding ambiguity issue is inherent in RGB-encoding used by SGS-SLAM. ActiveSGM maintains a global semantic distribution rather than relying on RGB features, which avoids this problem and leads to a better semantic understanding.
> We will clarify this in the revised version.
>
> ```
> Q1.10 [MP3D] Reasons for improved performance
> ```
>
> The improved performance comes from both the 3D Gaussian representation and the semantic exploration criterion — unlike baselines in Table 1 (blue) that use hard semantic assignments (each voxel can only have one class label), our soft semantic representation (distribution of most likely classes) combined with active view selection leads to better semantic and geometric reconstruction, as shown in Tables S.6 and S.7.
>
> ```
> Q1.11 Differences between baselines and ActiveSGM
> ```
>
> We refer the reviewer to Q1.2..
>
> ```
> Q1.12 Usage of ground-truth semantic labels
> ```
>
> We did not use ground-truth semantic labels in ActiveSGM, as our focus is on the more realistic scenario of handling noisy observations; instead, we aim to improve reconstruction performance using only imperfect pseudo labels.

---

> > ### Comment · Reviewer_fEfZ · 2025-08-07
> >
> > I have read the rebuttal and the comments with the other reviewers. Overall, while I appreciate that there are differences in the proposed to prior work - in particular, ActiveGAMER - those are not particularly large deviations from current SOTA. The results also show relatively minor performance improvement. Furthermore, the work is limited to simulated datasets, though I appreciate the difficulty of running real world experiments. In sum, I do not feel that for this top venue, the work, at this time, makes a sufficient contribution.

---

> > > ### Author Response · Authors · 2025-08-08
> > >
> > > We sincerely thank the reviewer for their feedback during the discussion phase. Building on the current discussion, we would like to highlight how ActiveSGM performs in the task of active semantic mapping, particularly on **MP3D**, a challenging large-scale real-world dataset with complex layouts and less common objects:
> > >
> > > - **Active semantic mapping** (Main Paper, Table 1 – red, all baselines are active semantic mapping methods): ActiveSGM achieves an mIoU of 65.58%, substantially surpassing the current SOTA Zhang et al. at 42.92%.
> > >
> > > - **Geometric reconstruction** (Supplement, Table S.6): ActiveSGM attains 1.56 cm Accuracy, 1.77 cm Completeness, and a 97.35% Completeness Ratio, outperforming ActiveGAMER (1.66 cm / 2.30 cm / 95.32%).
> > >
> > > - **Novel view synthesis** (Supplement, Table S.7): Even without dedicated photometric refinement, ActiveSGM reaches a PSNR of 26.16, exceeding ActiveGAMER (with refinement) at 24.76.
> > >
> > > These results consistently demonstrate ActiveSGM’s superiority across semantic, geometric, and photometric dimensions, reinforcing its effectiveness in complex, real-world active semantic mapping scenarios.
> > >
> > > We would like to repeat our previous comment that the standard for acceptance at top-tier venues does not require real-world experiments for active mapping, even without semantics which would further complicate such experiments.

---

> ### Author Response · Authors · 2025-08-05
>
> Dear Reviewer,
>
> Thank you again for your detailed feedback and for taking the time to review our submission. We hope our rebuttal has addressed your concerns clearly. If you have any additional questions or thoughts, we would greatly appreciate the opportunity to clarify further during the discussion period.
>
> Please let us know if there is any aspect of the paper or our response that you’d like us to elaborate on.
>
> Best regards,
>
> Authors

---

### Note · Authors · 2025-08-15

We sincerely thank all reviewers for their constructive feedback, insightful comments, and active participation during the discussion stage. We are encouraged that multiple reviewers recognized our work as **the first to address active semantic mapping with 3D Gaussian Splatting (3DGS) and to introduce a novel, memory-efficient sparse semantic representation**. Importantly, our method is the only active semantic mapping approach that does not rely on ground-truth semantics, instead using pseudo labels from a pre-trained model—making our approach **more robust to noisy semantic data and more practical for real-world deployment**.

As we are the first to tackle this problem, there are no direct baselines for a one-to-one comparison. Nevertheless, we carefully designed our experiments to highlight the unique contributions of ActiveSGM. If the paper is accepted, we will further improve the manuscript by clarifying the experimental settings, better explaining the purpose of each experiment, and incorporating the reviewers’ suggestions to strengthen the presentation and comparisons.

---

### Decision · Program_Chairs · 2025-09-17

**Decision:**

Accept (poster)

**Comment:**

The recommendations from the reviewers are: BR (fEfZ), BA (7urF), A (MRuN), A (f7pY)

Despite reminders, f7pY provided a superficial review and did not participate to the discussion. We had thus to discard this review.

The remaining concerns by fEfZ and 7urF is the similarity with ActiveGAMER.  However, the committee believes the authors answered correctly when discussing the introduction of the semantic information.
fEfZ is also (slightly) concerned by the lack of real world experiments.

MRuN is much more convinced about the contribution made by the paper and has only minor concerns.  Since the authors answered correctly about the concern on the similarity with ActiveGAMER in the view of the committee, the committee decided to propose to accept this submission.